# MIGA: Make Train-Free Infinite Frame Generation Great Again for Consistent Long Videos

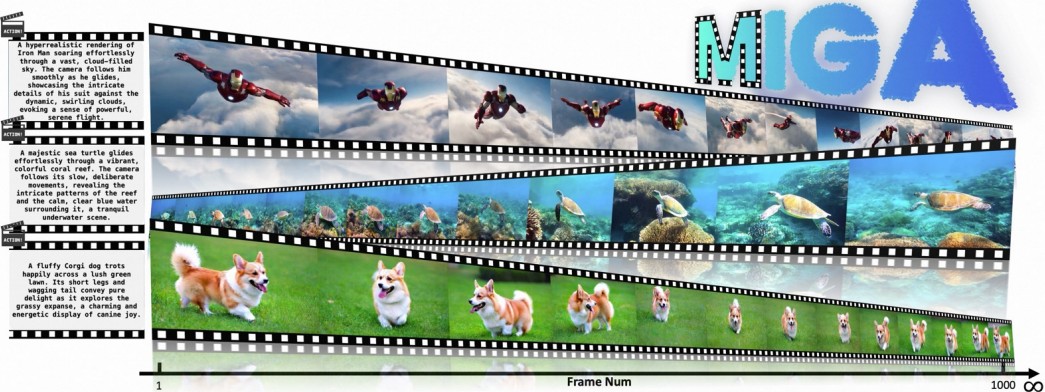

Figure 1: **MIGA enables temporally consistent, infinite-frame ($\infty$) video generation in a training-free manner.** We present three long videos (1000+ frames) generated by MIGA, while the foundation model used by MIGA, Wan2.1-1.3B (Wan et al., 2025), supports only 81 frames by default.

## Abstract

Without relying on significant computational or data resources, train-free long video generation aims to extend the duration of foundation video generation models, which are typically limited to short videos. Direct noise prediction on the entire long latents incurs substantial computational overhead. In contrast, frame-level autoregressive frameworks, *e.g.*, FIFO-diffusion, offer the advantage of generating infinitely long videos with constant memory consumption. However, the substantial gap between training and inference phases hinders the effective utilization of foundation models. Furthermore, maintaining long-term consistency is central to long video generation, yet existing methods pay insufficient attention to this aspect. To mitigate these concerns, we propose **MIGA**, a novel infinite-frame long video generation method. **(i)** Firstly, considering that the training-inference gap mainly stems from the excessive noise span of latents fed to the model during inference, we propose an effective two-stage alignment mechanism. By partitioning the generation process of existing frameworks into two dedicated stages with reduced noise spans, the capabilities of advanced foundation models are efficiently unlocked. **(ii)** Additionally, building upon the intrinsic properties of frame-level autoregressive frameworks, we introduce an innovative dual consistency enhancement mechanism. Specifically, our self-reflection approach evaluates and corrects early high-noise frames, while our long-range frame guidance approach leverages later low-noise frames with broad coverage to steer the generation process. These strategies jointly promote consistency in the generated content. **(iii)** Finally, extensive experiments on VBench and NarrLV demonstrate the state-of-the-art performance of MIGA.

## 1 Introduction

Recent advances in video generation (Wan et al., 2025; Kong et al., 2024; Yang et al., 2024) have demonstrated impressive capabilities in synthesizing short video clips. However, many real-world applications, such as film production, game development, and world simulation, require coherent long video generation (Cho et al., 2024). Building long video generation models from scratch typically requires substantial computational and data resources, owing to the inherent complexity of the video

modality (Waseem & Shahzad, 2024). Given the remarkable performance of off-the-shelf foundation video generation models on short videos (Chen et al., 2024b; Wan et al., 2025), a more efficient and practical approach is to extend their generation length in a training-free manner (Qiu et al., 2023).

To achieve train-free long video generation, one straightforward strategy is to increase the number of latents fed into foundation models and design specific mechanisms that transfer their short-term generation capabilities to long video scenarios. For example, FreeNoise (Qiu et al., 2023) ingeniously reorganizes the initial noise and further models temporal dependencies using window-based fusion. Following this paradigm, FreeLong (Lu et al., 2024) and FreePCA (Tan et al., 2025) facilitate the complementary integration of global and local information from the perspectives of frequency and principal component analysis, respectively. Although these methods have shown promising results, their memory requirements increase proportionally with the number of generated frames, which significantly restricts the achievable video length (*e.g.*, generating minute-long videos).

To enable infinite frame generation, recent studies such as Diffusion-Forcing (Chen et al., 2024a) and AR-Diffusion (Sun et al., 2025) attempt to assign different noise levels to different latent features, thereby empowering diffusion models to iteratively generate in an autoregressive fashion. Notably, FIFO-Diffusion (Kim et al., 2024) maintains a noise queue where noise levels increase sequentially along the frame dimension, and employs a first-in-first-out denoising process for frame-level autoregressive generation. More importantly, this approach requires only fixed memory consumption, enabling FIFO-Diffusion to support infinite-frame generation.

Despite these merits, train-free frame-level autoregressive models such as FIFO-Diffusion still leave considerable room for further improvement. On one hand, a substantial gap exists between training and inference in long video generation (Kim et al., 2024). In particular, during training, the model is exposed to input latents with a single noise level, whereas during inference, it must handle multiple noise levels corresponding to the number of frames. These discrepancies prevent the foundation generation models from fully realizing their potential, which in turn leads to issues such as content drift and visual artifacts (Cai et al., 2025). On the other hand, long-term consistency is a central objective for long video generation (Henschel et al., 2025; Yang et al., 2025), yet existing methods pay insufficient attention to this goal. For example, FIFO-Diffusion only facilitates feature interaction between neighboring chunks through lookahead denoising, but lacks explicit modeling of long-range frame dependencies, resulting in suboptimal long video quality.

To address these limitations, we propose **MIGA**, a novel train-free method for infinite-frame video generation. **(i)** First, we propose an intuitive and effective **two-stage training-inference alignment mechanism** to mitigate the inherent training-inference gap in existing train-free autoregressive frameworks. As this gap primarily arises from the excessive noise span of latents fed to the model during inference, we alleviate it through two dedicated optimization stages. The first stage maintains a zigzag-structured latent queue to proactively narrow the noise span of input latents. In the second stage, once all latents are denoised to the same noise level, a unified denoising process is conducted, achieving a noise span that matches that of the training phase. **(ii)** Furthermore, leveraging the properties of the maintained long latents queue, we present an innovative **dual consistency enhancement mechanism** to promote long-term consistency. For early high-noise latents, we design a self-reflection approach that efficiently evaluates and promptly corrects them, thereby ensuring consistency in the subsequently generated video. Unlike existing methods that rely on external evaluators and redundant computations (Yang et al., 2025; He et al., 2025), our approach achieves this solely through self-similarity analysis among early latents. For the later low-noise latents, we introduce a long-range frame guidance approach that incorporates them into each denoising iteration, facilitating feature interactions between distant frames. **(iii)** Benefiting from these improvements, MIGA achieves significant gains of 4.7% and 2.0% in subject and background consistency on VBench (Huang et al., 2024), respectively, compared to FIFO-Diffusion with a similar framework. Moreover, evaluations on NarrLV (Feng et al., 2025) demonstrate that MIGA exhibits exceptional capability in generating rich narrative content.

In summary, our contributions are as follows: **(i)** To inherit the merits of train-free frame-level autoregressive frameworks while alleviating their limitations in training-inference gap and long-term consistency modeling, we propose a novel infinite-frame generation method, MIGA. **(ii)** We design an effective two-stage training-inference alignment mechanism that proactively mitigates the training-inference gap by optimizing the noise span. Furthermore, we introduce an innovative dual consistency enhancement mechanism that promotes long-term consistency through self-reflection

and long-range frame guidance. **(iii)** Comprehensive experiments on the mainstream VBench and NarrLV benchmarks demonstrate that MIGA achieves new state-of-the-art performance.

## 2 RELATED WORKS

**Text-to-Video Generation.** Recently, the field of video generation has witnessed remarkable advancements. Early approaches primarily adopted frameworks that combine 2D spatial and 1D temporal modeling, such as VideoCrafter (Chen et al., 2023; 2024b), and Stable Video Diffusion (Blattmann et al., 2023). These have progressively transitioned into more advanced 3D full-attention architectures, as illustrated by Video Diffusion Models (Ho et al., 2022) and CogVideoX (Yang et al., 2024). Recently developed foundation models, including HunyuanVideo (Kong et al., 2024) and Wan (Wan et al., 2025) have further contributed to the improvement in video quality. Despite offering more accessible tools for video generation, current video diffusion models are generally constrained to training on short, fixed-length videos (Lu et al., 2024; Lu & Yang, 2025; Tan et al., 2025). Given the crucial role of long videos in practical scenarios (Cho et al., 2024), achieving consistent generation of long videos has emerged as a prominent research topic.

**Long Video Generation.** In pursuit of long video generation, several studies (Yan et al., 2025; Guo et al., 2025b; Teng et al., 2025; Chen et al., 2025; Xiao et al., 2025; Zhang & Agrawala, 2025; Deng et al., 2024) have introduced specialized model architectures and performed large-scale training on carefully curated datasets. Nevertheless, the heavy reliance on extensive computational and data resources makes these train-dependent approaches challenging for widespread adoption within the community (Lu et al., 2024). To address this, recent efforts have investigated train-free strategies for long video synthesis, aiming to efficiently extend the output duration of foundation video generators in a resource-friendly manner. For example, Gen-L-Video (Wang et al., 2023) extends video length by merging overlapping subsequences with a sliding-window method. FreeNoise (Qiu et al., 2023), FreeLong (Lu et al., 2024), and FreePCA (Tan et al., 2025) integrate local and global features by leveraging discovered patterns in initialization noise, frequency distributions, and principal component structures. RIFLEx (Zhao et al., 2025) further tackles periodic repetition in long video generation by refining temporal position encodings (Su et al., 2024). In contrast to the aforementioned methods, which are restricted to finite extensions, FIFO-Diffusion (Kim et al., 2024) endows diffusion models with frame-level autoregressive generation capabilities through an ingenious noise space design (Chen et al., 2024a; Liu et al., 2025b), thereby supporting infinite frame synthesis with a fixed memory cost. To retain the merits of this frame-level autoregressive approach, while mitigating its significant training-inference gap and limitations in consistency modeling, we propose a novel method—MIGA.

## 3 METHODS

### 3.1 PRELIMINARIES: TRAIN-FREE FRAME-LEVEL AUTOREGRESSIVE GENERATION.

Mainstream diffusion-based video generation models typically comprise a conditional encoder (*e.g.*, a text encoder), a variational autoencoder (VAE), and a noise prediction network $\varepsilon_\theta(\cdot)$. The VAE enables bidirectional mapping between pixel-level video data and compact latents, $\mathbf{z}_0 = [\mathbf{z}_0^1; ...; \mathbf{z}_0^f] \in \mathbb{R}^{f \times l \times d}$, where $f$, $l$, and $d$ represent the frame count, tokens per frame, and token dimension, respectively. For clarity, we regard the latent feature of each frame (*e.g.*, $\mathbf{z}_0^i \in \mathbb{R}^{l \times d}, i \in [1, f]$) as a basic unit throughout this paper. For example, the number of latents in $\mathbf{z}_0$ is $f$. Given a trained $\varepsilon_\theta(\cdot)$, the fully denoised latents $\mathbf{z}_0$ can be recovered from Gaussian noise $\mathbf{z}_T \sim \mathcal{N}(0, I)$. Following a time step schedule $0 = \tau_0 < \tau_1 < \cdots < \tau_T = T$, $\mathbf{z}_0$ is generated by progressively refining $\mathbf{z}_{\tau_T} = [\mathbf{z}_{\tau_T}^1; \ldots; \mathbf{z}_{\tau_T}^f]$ over $T$ steps with a sampler $\phi(\cdot)$ (*e.g.*, DDPM (Ho et al., 2020)). Each denoising step is formulated as:

$$[\mathbf{z}_{\tau_{t-1}}^1; \ldots; \mathbf{z}_{\tau_{t-1}}^f] = \phi([\mathbf{z}_{\tau_t}^1; \ldots; \mathbf{z}_{\tau_t}^f], [\tau_t; \ldots; \tau_t]; \varepsilon_\theta), \qquad (1)$$

where $\mathbf{z}_{\tau_t}^i$ denotes the latent of the $i$-th frame at time step $\tau_t$. For convenience, conditional inputs (*e.g.*, text prompts) are omitted in the above formulation.

To enable a foundation model that can only generate $f_0$ frames to produce long videos consisting of $N$ frames ($N \gg f_0$), frame-level autoregressive generation centers around maintaining a latents queue $\mathcal{Q} = \{\mathbf{z}_{\tau_1}^1; \ldots; \mathbf{z}_{\tau_T}^T\}$, which contains $T$ latents (*i.e.*, its length $L$ equals the total number of

(a) *FIFO−Diffusion*          (b) *Our TTA*

Figure 2: **Inference framework comparison between FIFO-Diffusion and our Two-Stage Training-Inference Alignment (TTA) mechanism. (a)** FIFO-Diffusion achieves frame-level autoregressive generation by maintaining a queue of latents with progressively increasing noise levels, resulting in an excessive noise span among the local latents fed to the model. **(b)** Our TTA effectively reduces the noise span: Stage 1 performs zigzag denoising by slowing down the rate of noise changes, and Stage 2 applies unified denoising once the output latents reach the same noise level.

denoising steps $T$) with progressively increasing noise level, as illustrated in Fig. 2 (a). After applying one inference step to all latents in $\mathcal{Q}$:

$$\{\mathbf{z}_{\tau_0}^1; \ldots; \mathbf{z}_{\tau_{T-1}}^T\} = \Phi(\{\mathbf{z}_{\tau_1}^1; \ldots; \mathbf{z}_{\tau_T}^T\}, \{\tau_1; \ldots; \tau_T\}; \varepsilon_\theta), \qquad (2)$$

the first latent in the queue, $\mathbf{z}_{\tau_0}^1$, becomes a fully denoised, clean latent. By dequeuing $\mathbf{z}_{\tau_0}^1$ from $\mathcal{Q}$ and appending a new Gaussian latent $\mathbf{z}_{\tau_T}^T$ to the end, the process can be repeated to realize frame-level autoregressive generation. Notably, since the queue length $T$ is typically greater than the number of frames $f_0$ that the model can process, a single inference step of the sampler $\Phi(\cdot)$ over $\mathcal{Q}$ involves multiple executions of the standard sampler $\phi(\cdot)$. For instance, FIFO-Diffusion employs a sliding window approach with a window size of $f_0$ and a stride of $\lfloor f_0/2 \rfloor$.

Prior to performing the above autoregressive generation, it is necessary to initialize $\mathcal{Q}$ appropriately. For details of initialization and the autoregressive generation procedure, please refer to App. A.1.1.

### 3.2 TWO-STAGE TRAINING-INFERENCE ALIGNMENT (TTA) MECHANISM.

The effectiveness of the aforementioned train-free frame-level autoregressive generation relies on the assumption that the foundation model can perform noise prediction on $f_0$ latents with varying noise levels. However, the model is trained to denoise $f_0$ latents at unified noise levels. This significant gap between training and inference hinders the foundation model's full generative potential. Although FIFO-Diffusion (Kim et al., 2024) has considered this issue and theoretically proved that the error introduced by train-free autoregressive generation is bounded by the span of noise levels, its final approach still requires the model to handle latents with a noise span of $f_0$. Given the impact of the noise span in input latents, a natural question emerges: *can we further reduce the noise span of latents fed to the model during inference, so as to better align the input condition with that of training?* Motivated by this, we decompose the generation process into two stages, aiming to maximally align training and inference by intuitively and effectively reducing the noise span.

**Stage 1: Zigzag Iterative Denoising.** Autoregressive generation inherently requires maintaining a noise queue that inevitably covers a range of noise levels (Chen et al., 2024a; Sun et al., 2025). To reduce the noise span of latents processed by the model, an intuitive adjustment is to slow down the rate at which noise levels change within the queue. Specifically, as shown in Fig. 2 (b), we initialize and maintain a noise queue $\mathcal{Q}_{s_1}$ as follows:

$$\mathcal{Q}_{s_1} = \{\underbrace{\mathbf{z}_{\tau_e}^1, \cdots, \mathbf{z}_{\tau_e}^{L_{zig}}}_{L_{zig}}, \underbrace{\mathbf{z}_{\tau_{e+1}}^{L_{zig}+1}, \cdots, \mathbf{z}_{\tau_{e+1}}^{2L_{zig}}}_{L_{zig}}, \cdots, \underbrace{\mathbf{z}_{\tau_T}^{L-L_{zig}+1}, \cdots, \mathbf{z}_{\tau_T}^L}_{L_{zig}}\}. \qquad (3)$$

Unlike existing methods that change the noise level with every single latent frame, our queue alters the noise level every $L_{zig}$ latents. This zigzag structure provides the model with a smoother noise span across inputs, contributing to mitigating the gap between training and inference. At each iteration, we dequeue the first $L_{zig}$ latents $\mathbf{z}_{\tau_e}^i$ (where $i \in [1, L_{zig}]$) from the front of the queue, and append $L_{zig}$ new Gaussian latents $\mathbf{z}_{\tau_T}^T$ to its end. It is important to note that the time step $\tau_e$ of the first $L_{zig}$ latents in the queue is greater than $\tau_0$, which means that Stage 1 only partially completes the denoising process. The subsequent denoising steps are carried out in Stage 2.

**Stage 2: Denoising at a Unified Noise Level.** After $n$ iterations in Stage 1, we obtain $nL_{\text{zig}}$ latents, all at the same time step $\tau_{e-1}$. These latents form the queue $\mathcal{Q}_{s_2}$ to be processed in the second stage:

$$\mathcal{Q}_{s_2} = \{\mathbf{z}_{\tau_{e-1}}^1, \mathbf{z}_{\tau_{e-1}}^2, \ldots, \mathbf{z}_{\tau_{e-1}}^{nL_{\text{zig}}}\}. \tag{4}$$

Since all latent frames have an identical noise level, the model only needs to process latents with the same noise intensity in each denoising operation. This setup aligns well with the conditions seen during training. After performing $(e-1)$ iterative denoising steps on $\mathcal{Q}_{s_2}$, we obtain $nL_{\text{zig}}$ fully denoised frames (*i.e.*, the desired number of frames in the generated video is $N = nL_{\text{zig}}$). The specific details of the two-stage process are provided in App. A.1.2.

### 3.3 DUAL CONSISTENCY ENHANCEMENT (DCE) MECHANISM.

Although the TTA mechanism effectively mitigates the gap between training and inference, it still lacks dedicated modeling designs for the crucial goal of long-term generation tasks, *i.e.*, maintaining long-term consistency. To address this issue, we propose an innovative dual consistency enhancement mechanism based on the characteristics of our maintained latent queue. Specifically, the self-reflection mechanism focuses on latents at the tail of the queue, efficiently evaluating and correcting newly added latents. Besides, the long-range frame guidance mechanism targets latents at the head of the queue, incorporating long-range, low-noise latents into each local denoising process. The roles of these two methods within the queue are shown in the framework diagram in Fig. A1 of App. A.1.3.

**Self-Reflection.** Recent advances in LLMs (Guo et al., 2025a; Jaech et al., 2024; Bai et al., 2023; Team et al., 2025) have explored test-time scaling (TTS) (Zhang et al., 2025), which can significantly improve response quality by allocating additional computation during inference. Inspired by these advances, researchers have sought to transfer TTS techniques to video generation tasks (Ma et al., 2025; Liu et al., 2025a; He et al., 2025; Singhal et al., 2025), introducing multiple candidate latents

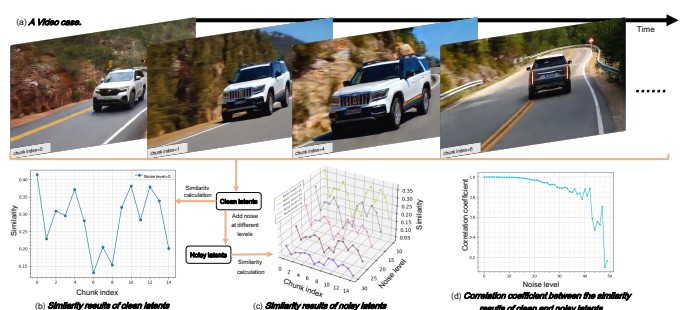

Figure 3: **Modeling insight behind our self-reflection mechanism.** (a) A video case containing the consistency anomaly. (b-d) Similarity computation between clean and noisy latents, along with the corresponding correlation coefficient analysis results.

and employing specific evaluation and selection strategies to achieve higher-quality videos. In contrast to existing methods that generally focus on fixed-length video generation, our self-reflection mechanism effectively integrates the idea of TTS with the characteristics of frame-level autoregressive generation for long videos. Given that temporal consistency is a crucial modeling aspect in long video generation tasks, long-term consistency is typically set as the primary objective when extending the search time. The most relevant prior work is ScalingNoise (Yang et al., 2025), which employs a consistency reward function to guide long video generation. The differences between our approach and ScalingNoise, as well as other TTS methods, are discussed in detail below.

Our self-reflection approach interprets TTS as comprising two processes: first, adaptively evaluating the locations where anomalies (*e.g.*, abrupt drops in consistency) occur; second, performing an expanded search at these points for correction (please refer to Fig. A1 (b) for the flowchart). Unlike previous methods that either conduct search at every step (Yang et al., 2025) or at predefined scheduler steps (He et al., 2025), our approach aims to efficiently and flexibly determine when to trigger expanded search. To achieve this, a straightforward and accurate consistency metric is required. Existing methods often rely on external models for quantitative assessment. For example, ScalingNoise uses DINO (Caron et al., 2021) for consistency evaluation, introducing redundancy into the pipeline. Moreover, since these models require clean pixel inputs, consistency assessment during intermediate denoising steps necessitates additional denoising and VAE decoding procedures (Yang et al., 2025), resulting in high computational overhead. To overcome these limitations, we propose a more efficient consistency evaluation strategy inspired by the following observation.

Firstly, the latent space produced by the VAE, after large-scale pre-training, exhibits strong interpretability (Kingma & Welling, 2013; Wang et al., 2024). Specifically, the distance between latents reflects the degree of difference between the corresponding video frames. Therefore, we can leverage the cosine similarity between different latents as a consistency metric, thereby avoiding the need for additional external evaluation models. Formally, let $\mathbf{q}_{\text{eval}} \in \mathbb{R}^{f_{\text{eval}} \times l \times c}$ denote the $f_{\text{eval}}$ consecutive latents to be evaluated, and $\mathbf{q}_{\text{ref}} \in \mathbb{R}^{f_{\text{ref}} \times l \times c}$ denote those of the preceding $f_{\text{ref}}$ adjacent latents. The consistency score $C_{\text{score}}$ is computed as follows:

$$\mathbf{q}'_{\text{eval}} = \text{norm}_1 \left( \text{mean}_2(\mathbf{q}_{\text{eval}}) \right), \quad \mathbf{q}'_{\text{ref}} = \text{norm}_1 \left( \text{mean}_2(\mathbf{q}_{\text{ref}}) \right), \tag{5}$$

$$C_{\text{score}} = \text{mean}_1 \left( \text{mean}_2 \left( \mathbf{q}'_{\text{eval}} {\mathbf{q}'_{\text{ref}}}^{\text{T}} \right) \right), \tag{6}$$

where $\text{norm}_i(\cdot)$ and $\text{mean}_i(\cdot)$ denote the normalization and mean operations along the $i$-th dimension ($i \geq 0$), respectively, and $T$ denotes the matrix transpose. Fig. 3 (a) and (b) illustrate the results of sequentially measuring consistency over video segments ($f_{\text{eval}} = 4$, $f_{\text{ref}} = 8$) in a clean video containing the consistency anomaly. It is evident that the proposed metric can effectively identify the position of the consistency disruption. However, considering the influence of the initial noisy latents (Qiu et al., 2023) on the final generated content—such as the overall layout of the video being largely determined in the early denoising stages—we aim to assess consistency at the early high-noise latents rather than at the clean latents, in order to enable timely adjustments. A straightforward solution is to fully denoise these high-noise latents and then compute $C_{\text{score}}$. Undoubtedly, such frequent denoising operations would incur substantial computational overhead. Fortunately, we observe that the early high-noise latents and the final clean latents exhibit a strong correlation in terms of $C_{\text{score}}$. As shown in Fig. 3 (c), higher noise levels reduce the absolute magnitude of the $C_{\text{score}}$ curve, yet the fluctuation patterns remain similar across different noise levels. Fig. 3 (d) presents the correlation coefficients between the $C_{\text{score}}$ curves under various noise levels and those of clean latents, indicating strong correlation even at higher noise intensities (*e.g.*, 40, with the maximum noise level being 50).

Leveraging this insight, we can timely evaluate consistency at early stages. Specifically, for the latent queue $\mathcal{Q}$ covering all noise levels, we define a judgment index $f_{\text{judg}}$ at its tail (*i.e.*, the early high-noise latents). During each iteration over $\mathcal{Q}$, the latents in $[f_{\text{judg}} - f_{\text{ref}}, f_{\text{judg}} - 1]$ are used as reference to evaluate the latents in $[f_{\text{judg}}, f_{\text{judg}} + f_{\text{eval}} - 1]$. When the decrease in $C_{\text{score}}$ between adjacent chunks exceeds the threshold $\delta_{\text{adju}}$, an expanded search is triggered for further correction.

Upon detecting a consistency anomaly, all latents after the position $f_{\text{judg}}$, $q_{\text{samp}}^{\text{init}} = \{z^i\}_{i=1}^{L-f_{\text{judg}}+1}$, are considered for correction (*i.e.*, the required format for our search samples). Benefiting from our early determination of the judgment node, the number of latents included in the search sample is relatively small. Taking into account the diversity and effectiveness of the search samples, we design a progressively guided search strategy. Assuming the number of search samples is $n_{\text{samp}}$, we first randomly sample $n_{\text{samp}}$ Gaussian noise latents as starting points for each sample: $q_{\text{samp}}^k = \{z^1\}$, $k \in [1, n_{\text{samp}}]$. Subsequently, we use the preceding $f_{\text{guid}}$ latents before $f_{\text{judg}}$, which have passed our evaluation, as the guiding information, denoted as $q_{\text{guid}} = \{z^i\}_{i=1}^{f_{\text{guid}}}$. For each sample $q_{\text{samp}}^k$, we concatenate it with $q_{\text{guid}}$ and perform iterative denoising. After each iteration, only the latents in $q_{\text{samp}}^k$ are updated, and a new noise frame is appended to $q_{\text{samp}}^k$. After $L - f_{\text{judg}}$ iterations, $q_{\text{samp}}^k$ contains $(L - f_{\text{judg}} + 1)$ latents, maintaining the same data format as $q_{\text{samp}}^{init}$.

After obtaining $k$ candidate samples, the consistency score $C_{\text{score}}^k$ is computed for each candidate based on its first $f_{\text{eval}}$ latents. If the highest score among them exceeds $C_{\text{score}}^{\text{init}}$, we replace $Q_{\text{samp}}^{\text{init}}$ with the $Q_{\text{samp}}^k$ corresponding to the highest score, thereby completing the correction process. For details on the sampling and correction procedure, please refer to Alg. 6 in App. A.1.3.

**Long-Range Frame Guidance.** When applying the foundation model for sliding inference on the maintained queue $\mathcal{Q} = \{z^i\}_{i=1}^L$, the model can consider only $f_0$ nearby latents during each denoising operation. To enable interactions among distant latents and thereby enhance the consistency of generated videos, we propose a simple yet effective long-range frame guidance method. Specifically, when the model processes latents within a local window, we explicitly sample $m_{\text{guid}}$ latents sparsely from earlier positions in $\mathcal{Q}$. Since these latents are relatively clean, we utilize them to guide the denoising of the current local latents (please refer to Fig. A1 (a) for the flowchart). Formally, when

Figure 4: **Illustration of ablation study results.** (a-d) Starting from the *baseline*, our *Stage 1*, *Stage 2*, and *DCE* mechanism are sequentially added. Yellow bboxes in the first frame indicate regions with prominent noise. Red bboxes denote regions in the current frame where the subject exhibits noticeable inconsistency compared to previous frames. Better viewed in color with zoom-in.

the model slides to position $l$ ($l \in [1, L - m_{\text{guid}}]$) for inference, the input $q_{\text{input}}$ is defined as follows:

$$q_{\text{input}} = \begin{cases} [z^l, z^{l+1}, \ldots, z^{l+f_0-1}], & l \le m_{\text{guid}} \\ [z^1, z^2, \ldots, z^{m_{\text{guid}}}, z^l, z^{l+1}, \ldots, z^{l+f_0-m_{\text{guid}}-1}], & m_{\text{guid}} < l \le L - m_{\text{guid}} \end{cases} \tag{7}$$

for the head latents in the queue (*i.e.*, $l \le m_{\text{guid}}$), we do not apply long-range guidance due to the insufficient number of preceding latents. It should be noted that these latents are also obtained by iterative denoising propagated from the tail of the sequence, such that they have also been guided by their preceding latents during the process. For the selection of $m_{\text{guid}}$ guidance latents, we uniformly sample $m_{\text{guid}}$ latents from the $\min(m_{\text{guid}} L_{\text{zig}}, l - 1)$ prior latents before position $l$.

# 4 EXPERIMENTS

## 4.1 IMPLEMENTATION DETAILS.

**Model Implementation.** To ensure fair comparison with existing training-free long video generation methods, we apply MIGA to the widely-used foundation model, VideoCrafter2 (Chen et al., 2024b). Besides, we incorporate MIGA into the latest model, Wan2.1-1.3B (Wan et al., 2025). By default, these models support generating 16 and 21 latents, respectively. For infinite-frame generation, the configuration of VideoCrafter2-based MIGA is as follows: $T = 64$, $L_{\text{zig}} = 4$, $\tau_e = 10$, $\delta_{\text{adju}} = 0.01$, and $m_{\text{guid}} = 6$. For Wan2.1-based MIGA, the configuration is $T = 54$, $L_{\text{zig}} = 7$, $\tau_e = 10$, $\delta_{\text{adju}} = 0.01$, and $m_{\text{guid}} = 4$. Moreover, thanks to the flexibility of the frame-level autoregressive framework, we can achieve multi-text control by providing different text conditions to the latents at various temporal positions. For more implementation details, see App. A.2 and App. A.3.

Table 1: Quantitative results of MIGA and baselines on VBench. The best results are highlighted in **bold**.

| Method | Infinite | S.C. | B.C. | M.S. | T.F. | O.S. |
|---|---|---|---|---|---|---|
| *VideoCrafter2-Based* | | | | | | |
| FreePCA | ✗ | 93.57 | 95.24 | 93.73 | 91.27 | 93.45 |
| FreeLong | ✗ | 95.72 | 96.42 | 98.38 | 97.28 | 96.95 |
| FIFO-Diffusion | ✓ | 92.92 | 95.01 | 97.19 | 94.94 | 95.02 |
| ScalingNoise | ✓ | 94.29 | 95.52 | 97.86 | 96.12 | 95.95 |
| MIGA (ours) | ✓ | **97.66** | **96.99** | **98.60** | **98.03** | **97.82** |
| *Wan2.1-Based* | | | | | | |
| FIFO-Diffusion | ✓ | 92.67 | 93.37 | 98.03 | 97.09 | 95.29 |
| MIGA (ours) | ✓ | **96.46** | **95.50** | **98.85** | **98.14** | **97.24** |

**Evaluation Benchmarks.** Our evaluations are conducted on the VBench (Huang et al., 2024) and NarrLV (Feng et al., 2025) benchmarks. Following the evaluation protocols of existing long video generators, we primarily use the video quality dimensions assessed by the VBench-Long toolkit. The commonly used metrics include subject consistency (S.C.), background consistency (B.C.), motion smoothness (M.S.), temporal flicker (T.F.), and their mean, Overall Score (O.S.). NarrLV is a recent benchmark for narrative expressiveness in long video models. We use evaluation prompts with Temporal Narrative Atom (TNA) counts of 2, 3, and 4, and report results on three dimensions, *i.e.*, scene attributes ($s_{\text{att}}$), target attributes ($t_{\text{att}}$), and target actions ($t_{\text{act}}$).

Table 2: Quantitative results of MIGA and baselines under varying TNA settings on NarrLV.

| Method | Infinite | TNA=2 | | | TNA=3 | | | TNA=4 | | |
|---|---|---|---|---|---|---|---|---|---|---|
| | | $s_{\text{att}}$ | $t_{\text{att}}$ | $t_{\text{act}}$ | $s_{\text{att}}$ | $t_{\text{att}}$ | $t_{\text{act}}$ | $s_{\text{att}}$ | $t_{\text{att}}$ | $t_{\text{act}}$ |
| *VideoCrafter2-Based* | | | | | | | | | | |
| FreePCA | ✗ | 56.96 | 58.72 | 56.41 | 53.61 | 53.93 | 52.57 | 50.46 | 57.28 | 53.27 |
| FreeLong | ✗ | 59.43 | 59.57 | 55.95 | 56.57 | 59.82 | 56.57 | 54.13 | 60.53 | 54.13 |
| ScalingNoise | ✓ | 59.28 | 55.47 | 58.09 | 53.27 | 58.14 | 54.05 | 52.37 | 58.41 | 53.59 |
| FIFO-Diffusion | ✓ | 67.02 | 63.55 | 58.29 | 61.15 | 60.64 | 58.42 | 66.09 | 66.01 | 54.66 |
| MIGA (ours) | ✓ | **69.78** | **63.94** | **59.01** | **63.53** | **61.05** | **59.52** | **68.87** | **68.77** | **55.78** |
| *Wan2.1-Based* | | | | | | | | | | |
| FIFO-Diffusion | ✓ | 67.77 | 64.25 | 65.40 | 55.42 | 59.02 | 58.91 | 57.43 | 56.10 | 53.89 |
| MIGA (ours) | ✓ | **79.32** | **67.87** | **67.94** | **69.48** | **66.33** | **63.86** | **75.05** | **72.31** | **62.90** |

**Compared Baselines.** For methods that extend video length by increasing input latents, we compare against FreePCA (Tan et al., 2025) and FreeLong (Lu et al., 2024). For approaches supporting infinite-length video generation, we select FIFO-Diffusion (Kim et al., 2024) and ScalingNoise (Yang et al., 2025) as representative baselines.

## 4.2 COMPARISON WITH BASELINES.

**VBench.** As shown in Tab. 1, MIGA achieves state-of-the-art performance across all metrics, for both foundation models. For the VideoCrafter2-based models, we standardized the generation length to 128 frames to ensure fair comparison. Compared to the strong baseline FreeLong, MIGA demonstrates further improvement in both subject and background consistency, which validates the effectiveness of our approach in enhancing video consistency. For the Wan2.1-based models, we evaluate the generation of 161-frame videos. Compared against FIFO-Diffusion, which also adopts the autoregressive framework, MIGA achieves comprehensive improvements in all metrics, highlighting the efficacy of our optimizations to the framework. It is worth noting that the consistency scores of Wan2.1-based MIGA are slightly lower than those of VideoCrafter2-based MIGA. We conjecture that this is because the latter primarily generates animation-style videos, where maintaining long-term consistency is relatively easier than in the realistic style videos produced by the former. A qualitative example and analysis are provided in App. B.2. Subsequent evaluation results on the NarrLV demonstrate that Wan2.1-based MIGA achieves notable advantages in narrative expressiveness.

**NarrLV.** To generate videos that correspond to the rich narrative content in NarrLV, we adopt a sequence of changing prompts guidance strategy similar to FIFO-Diffusion. Specifically, latents at different temporal positions are conditioned on distinct segments of the prompt. As shown in Tab. 2, FIFO-Diffusion outperforms existing VideoCrafter2-based methods in narrative expressiveness. Furthermore, our MIGA achieves additional improvements, which can be attributed to its design that enables more stable video content generation and thereby supports richer semantic expression.

**Qualitative Results.** Fig. 1 presents three 1k-frame videos generated by the Wan2.1-based MIGA, which closely follow the text prompts and maintain strong subject and background consistency. For additional visualizations, please refer to App. B.4.

## 4.3 ABLATION STUDY.

To explore the effectiveness of our proposed mechanism design, we conduct comprehensive ablation studies on the VideoCrafter2-based MIGA using VBench. Detailed implementation information for each setting can be found in App. B.1.

**Study on Core Mechanisms Designs.** The core contribution of our work is the introduction of the novel Two-Stage Training-Inference Alignment (TTA) and Dual Consistency Enhancement (DCE) mechanisms. As shown in Tab. 3, FIFO-Diffusion serves as the baseline without our proposed mechanisms. Introducing TIA and DCE individually yields overall score improvements of 2.03% and 1.73%, respectively,

Table 3: Ablation results of core mechanisms.

| TTA | DCE | S.C. | B.C. | M.S. | T.F. | O.S. |
|---|---|---|---|---|---|---|
| | | 92.92 | 95.01 | 97.19 | 94.94 | 95.02 |
| ✓ | | 96.74 | 96.75 | 97.57 | 97.12 | 97.05 |
| | ✓ | 96.10 | 96.47 | 97.88 | 96.56 | 96.75 |
| ✓ | ✓ | 97.66 | 96.99 | 98.60 | 98.03 | 97.82 |

demonstrating the effectiveness of our core mechanisms. Furthermore, combining these two mechanisms provides complementary benefits and further enhances performance.

Table 4: Ablation results of $L_{\text{zig}}$.

| $L_{\text{zig}}$ | S.C. | B.C. | M.S. | T.F. | O.S. |
|---|---|---|---|---|---|
| 1 | 94.23 | 94.52 | 97.98 | 96.47 | 95.80 |
| 2 | 94.24 | 95.93 | 98.55 | 97.90 | 96.66 |
| 4 | 95.37 | 95.96 | 98.65 | 98.02 | 97.00 |
| 6 | 95.14 | 96.04 | 98.60 | 97.97 | 96.94 |
| 8 | 95.54 | 95.96 | 98.56 | 97.90 | 96.99 |

Table 5: Ablation results of $m_{\text{guid}}$.

| $m_{\text{guid}}$ | S.C. | B.C. | M.S. | T.F. | O.S. |
|---|---|---|---|---|---|
| 0 | 94.23 | 94.52 | 97.98 | 96.47 | 95.80 |
| 2 | 94.66 | 94.72 | 98.64 | 98.05 | 96.52 |
| 4 | 94.59 | 94.58 | 98.64 | 98.10 | 96.48 |
| 6 | 95.45 | 95.69 | 98.45 | 97.89 | 96.87 |
| 8 | 95.32 | 95.12 | 98.60 | 98.05 | 96.77 |

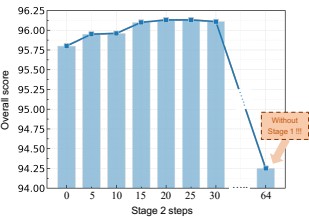

Figure 5: Ablation study on the steps in stage 2.

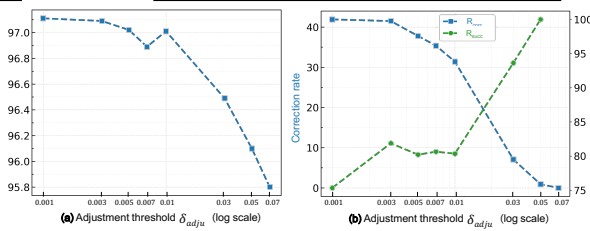

Figure 6: Ablation study on the adjustment threshold $\delta_{\text{adju}}$. **(a-b)** Effects of $\delta_{\text{adju}}$ on O.S., $R_{\text{corr}}$, and $R_{\text{succ}}$.

**Study on TTA.** Our TTA mechanism comprises two stages: stage 1 employs zigzag iterative denoising, and stage 2 applies denoising at a unified noise level. As shown in Tab. 6, sequentially introducing these two stages leads to notable performance improvements. As shown in Fig. 4 (a–c), stage 1 significantly reduces drastic anomalies present in the base-

Table 6: Ablation results of TTA design.

| Setting | S.C. | B.C. | M.S. | T.F. | O.S. |
|---|---|---|---|---|---|
| Baseline | 92.92 | 95.01 | 97.19 | 94.94 | 95.02 |
| +Stage 1 | 95.98 | 96.54 | 97.57 | 97.03 | 96.78 |
| +Stage 2 | 96.74 | 96.75 | 97.57 | 97.12 | 97.05 |

line, while stage 2 plays a key role in suppressing video noise. Furthermore, Tab. 4 shows the impact of the zigzag width $L_{\text{zig}}$ in stage 1 on model performance. As $L_{\text{zig}}$ increases, O.S. initially improves and then stabilizes. Thus, we choose $L_{\text{zig}} = 4$. Additionally, Fig. 5 illustrates the effect of varying the number of second-stage steps, *i.e.*, $(e - 1)$. As the step count increases, performance gains gradually stabilize. However, if only the second stage is performed, *i.e.*, inference is conducted directly on $N$ noise latents, model performance drops sharply. This approach is no longer autoregressive, as it requires simultaneous processing of independently initialized latents. In contrast, the omitted stage 1 leverages autoregressive generation to establish implicit information transfer among the initialized latents. Further discussion is provided in App. B.1.

**Study on DCE.** Our DCE mechanism consists of self-reflection and long-range frame guidance. Fig. 6 shows the impact of different adjustment thresholds $\delta_{\text{adju}}$, where a smaller $\delta_{\text{adju}}$ leads to more frequent searches. Fig. 6 (b) presents two metrics that reflect the search effectiveness. We denote the total inference steps, steps evaluated for correction, and steps actually corrected as $n_{\text{all}}$, $n_{\text{eval}}$, and $n_{\text{corr}}$, respectively. The correction rate and success rate are defined as $R_{\text{corr}} = n_{\text{corr}}/n_{\text{all}}$ and $R_{\text{succ}} = n_{\text{corr}}/n_{\text{eval}}$. As the threshold decreases, the model tends to perform broader searches, increasing the corrected steps (*i.e.*, $R_{\text{corr}}$) and consequently improving the O.S. metric, demonstrating test-time scaling capability. Since the number of steps with consistency anomalies within $n_{\text{all}}$ is limited, performance gains eventually converge. Further reducing the threshold increases $n_{\text{eval}}$ but not $n_{\text{corr}}$, leading to a lower $R_{\text{succ}}$. Balancing performance and computational cost, we set the default $\delta_{\text{adju}}$ to 0.01. Tab. 5 presents the effect of varying the number of guidance frames, with optimal performance achieved at $m_{\text{guid}} = 6$. App. B.1 provides an analysis of computational efficiency.

## 5 CONCLUSION

Train-free frame-level autoregressive generation frameworks, represented by FIFO-Diffusion, demonstrate the capability to generate infinitely long videos with constant memory cost. To build on these merits while overcoming limitations in the training-inference gap and long-term consistency, we introduced MIGA, a novel infinite-frame generation method. An effective two-stage training-inference alignment mechanism is developed to proactively mitigate the training-inference gap by optimizing the noise span. Additionally, an innovative dual consistency enhancement mechanism is proposed to improve long-term consistency through self-reflection and long-range frame guidance. Experiments show that MIGA achieves state-of-the-art results on VBench and NarrLV compared to existing train-free methods, demonstrating its ability to generate long videos with strong temporal consistency and rich narratives.

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

APPENDIX

# A  FURTHER DETAILS ON OUR METHOD

## A.1  PSEUDOCODE IMPLEMENTATION

In Sec. 3, we first take FIFO-Diffusion as an example to illustrate the concrete implementation process of the train-free frame-level autoregressive generation framework (see Sec. 3.1). Next, we analyze the significant gap between training and inference present in the existing framework, thereby motivating our **Two-Stage Training-Inference Alignment (TTA) Mechanism** (see Sec. 3.2). Finally, in light of the critical importance of consistency preservation in long video generation, our proposed **Dual Consistency Enhancement (DCE) Mechanism** is described in detail (see Sec. 3.3). In this section, we provide the detailed pseudocode implementations of each method to more clearly illustrate their procedures.

### A.1.1  PRELIMINARIES: TRAIN-FREE FRAME-LEVEL AUTOREGRESSIVE GENERATION.

As described in Sec. 3.1, existing frame-level autoregressive generation methods rely on maintaining a noise queue $\mathcal{Q} = \{\mathbf{z}_{\tau_1}^1; \ldots; \mathbf{z}_{\tau_T}^T\}$ with progressively increasing noise levels. Here, we take FIFO-Diffusion (Kim et al., 2024) as an example to illustrate the computation process.

The queue is first initialized, typically using clean initial latents generated by the foundation model. Since the queue length equals the denoising step count ($T$), which exceeds the initial latent count $f_0$, FIFO-Diffusion (see Alg. 1) uses the initial latents to initialize the last $f_0$ queue latents; the sampler (Ho et al., 2020; Lipman et al., 2022) then adds noise from $\tau_i$ to $\tau_j$ (where $j > i$ )according to the scheduler-specified noise strengths:

$$\mathbf{z}_{\tau_j} \leftarrow \Phi.\text{add\_noise}(\mathbf{z}_{\tau_i}, \tau_i, \tau_j). \tag{A1}$$

Next, the remaining $(T - f)$ latents are initialized by injecting varying levels of noise into the first latent.

After initializing the noise queue, iterative inference can be performed as specified in Eq. 1. As shown in Alg. 2, with each inference step, the noise level of all latents in the queue decreases by one, making the first latent clean; this latent is then dequeued from the queue and saved. To ensure continuous generation, a Gaussian noise latent is enqueued at the end of the queue at each step.

It is important to note that in Eq. 1, one inference step by the sampler over the entire queue requires multiple noise predictions from the foundation model $\epsilon_\theta(\cdot)$, since $\epsilon_\theta(\cdot)$ can only process $f_0$ latents at a time while the queue contains $T$ latents ($T > f$). Specifically, FIFO-Diffusion adopts a sliding window approach: as shown in Alg.3, the foundation model iteratively predicts noise for the queue using a window of size $f_0$ with a stride of $l_{\text{stride}}$.

### A.1.2  TWO-STAGE TRAINING-INFERENCE ALIGNMENT MECHANISM.

As discussed in Sec. 3.2, our Two-Stage Training-Inference Alignment (TTA) mechanism aims to mitigate the gap between training and inference by reducing the noise span of latents fed to the model during inference. In this section, we present the detailed implementation of TTA with pseudocode.

First, we need to initialize the latents queue. To ensure compatibility with the Zigzag Iterative Denoising in Stage 1, the adjacent $L_{\text{zig}}$ latents are assigned the same noise level during initialization. A notable difference from the initialization in FIFO-Diffusion is that, after initializing the queue tail with clean latents, we progressively guide the noise latent using these latents to complete the initialization of the entire queue, rather than simply duplicating semantically identical frames. The detailed initialization procedure is presented in Alg. 4.

Next, we perform our designed two-stage iterative generation process, as illustrated in Alg. 5. In Stage 1 (Zigzag Iterative Denoising), after each inference on the queue, we dequeue the $L_{\text{zig}}$ partially denoised latents from the head of the queue and save them, while $L_{\text{zig}}$ Gaussian noise latents are enqueued at the tail. After $N$ latents have been dequeued, Stage 2 performs the remaining denoising steps on these latents with identical noise levels.

---

**Algorithm 1** Initial latents construction in FIFO-Diffusion

---

**Require:** Denoising step count $T$, initial latents count $f_0$, denoising network $\epsilon_\theta(\cdot)$, and sampler $\Phi(\cdot)$

1: **Input**: initial clean latents $q_{\text{clean}} = \{\mathbf{z}_{\tau_0}^1; \ldots; \mathbf{z}_{\tau_0}^f\}$, time steps $t_{\text{clean}} = \underbrace{\{\tau_0; \ldots; \tau_0\}}_{f_0}$

2: **Output**: $\mathcal{Q}_{\text{init}} = \{\mathbf{z}_{\tau_1}^1; \ldots; \mathbf{z}_{\tau_T}^T\}$, $t_{\text{init}} = \underbrace{\{\tau_0; \ldots; \tau_T\}}_{T}$

3: $\mathcal{Q}_{\text{init}} = \{\}, \quad t_{\text{init}} = \{\}$
4: # Initialize the early latents using the first frame.
5: **for** $i = 1$ to $T - f_0$ **do**
6:     $\mathbf{z}_{\tau_i}^i \leftarrow \Phi.\text{add\_noise}(q_{\text{clean}}[0], t_{\text{clean}}[0], \tau_i)$
7:     $\mathcal{Q}_{\text{init}}.\text{enqueue}(\mathbf{z}_{\tau_i}^i)$
8:     $t_{\text{init}}.\text{enqueue}(\tau_i)$
9: **end for**
10: # Manually add noise to the initial latents.
11: **for** $i = 1$ to $f_0$ **do**
12:     $\mathbf{z}_{\tau_{T-f_0+i}}^i \leftarrow \Phi.\text{add\_noise}(q_{\text{clean}}[i], t_{\text{clean}}[i], \tau_{T-f_0+i})$
13:     $\mathcal{Q}_{\text{init}}.\text{enqueue}(\mathbf{z}_{\tau_{T-f_0+i}}^i)$
14:     $t_{\text{init}}.\text{enqueue}(\tau_{T-f_0+i})$
15: **end for**
16: **return** $\mathcal{Q}_{\text{init}}, \quad t_{\text{init}}$

---

**Algorithm 2** Frame-level autoregressive generation in FIFO-Diffusion

---

**Require:** Denoising step count $T$, generation latents count $N$, denoising network $\epsilon_\theta(\cdot)$, and sampler $\Phi(\cdot)$

1: **Input**: $\mathcal{Q} = \{\mathbf{z}_{\tau_1}^1; \ldots; \mathbf{z}_{\tau_T}^T\}$, $t = \underbrace{\{\tau_0; \ldots; \tau_T\}}_{T}$

2: **Output**: $\mathcal{Q}_{\text{gen}} = \{\mathbf{z}_{\tau_0}^1; \ldots; \mathbf{z}_{\tau_0}^N\}$
3: $\mathcal{Q}_{\text{gen}} = \{\}$
4: # Frame-level autoregressive generation.
5: **for** $i = 1$ to $N$ **do**
6:     $Q \leftarrow \Phi(Q, t; \epsilon_\theta)$
7:     $\mathbf{z}_{\tau_0}^i \leftarrow \mathcal{Q}.\text{dequeue}()$
8:     $\mathcal{Q}_{\text{gen}}.\text{enqueue}(\mathbf{z}_{\tau_0}^i)$
9:     $\mathbf{z}_{\tau_T}^T \sim \mathcal{N}(0, \mathbf{I})$
10:     $\mathcal{Q}.\text{enqueue}(\mathbf{z}_{\tau_T}^T)$
11: **end for**
12: **return** $\mathcal{Q}_{\text{gen}}$

---

---

**Algorithm 3** One-step inference over the latents queue in FIFO-Diffusion

---

**Require:** Denoising network $\epsilon_\theta(\cdot)$, initial latents count $f_0$, sliding window stride $l_{\text{stride}}$, and initial sampler $\phi(\cdot)$

1: **Input**: $\mathcal{Q} = \{\mathbf{z}_{\tau_1}^1; \ldots; \mathbf{z}_{\tau_T}^T\}$, $t = \underbrace{\{\tau_0; \ldots; \tau_T\}}_{T}$

2: **Output**: $\mathcal{Q}_{\text{gen}} = \{\mathbf{z}_{\tau_0}^1; \ldots; \mathbf{z}_{\tau_{T-1}}^T\}$

3: # Queue length equals denoising steps $T$.

4: $l_{\text{stride}} = [f_0/2]$

5: $n_{\text{iter}} = \lceil (T - f_0)/l_{\text{stride}} \rceil + 1$

6: **for** $i = 1$ to $n_{\text{iter}}$ **do**

7:     **if** $i < n_{\text{iter}}$ **then**

8:         $s_{\text{index}} = (i - 1) \times l_{\text{stride}} + 1$

9:         $e_{\text{index}} = s_{\text{index}} + f_0$

10:        $Q_{\text{temp}} \leftarrow \phi(Q[s_{\text{index}} : e_{\text{index}}], t[s_{\text{index}} : e_{\text{index}}]; \epsilon_\theta)$

11:        **for** $j = 1$ to $l_{\text{stride}}$ **do**

12:           $Q[s_{\text{index}} + j] = Q_{\text{temp}}[j]$

13:        **end for**

14:     **else**

15:        $s_{\text{index}} = T - f_0$

16:        $e_{\text{index}} = T$

17:        $Q_{\text{temp}} \leftarrow \phi(Q[s_{\text{index}} : e_{\text{index}}], t[s_{\text{index}} : e_{\text{index}}]; \epsilon_\theta)$

18:        **for** $j = 1$ to $f_0$ **do**

19:           $Q[s_{\text{index}} + j] = Q_{\text{temp}}[j]$

20:        **end for**

21:     **end if**

22: **end for**

23: $\mathcal{Q}_{\text{gen}} = \mathcal{Q}$

24: **return** $\mathcal{Q}_{\text{gen}}$

---

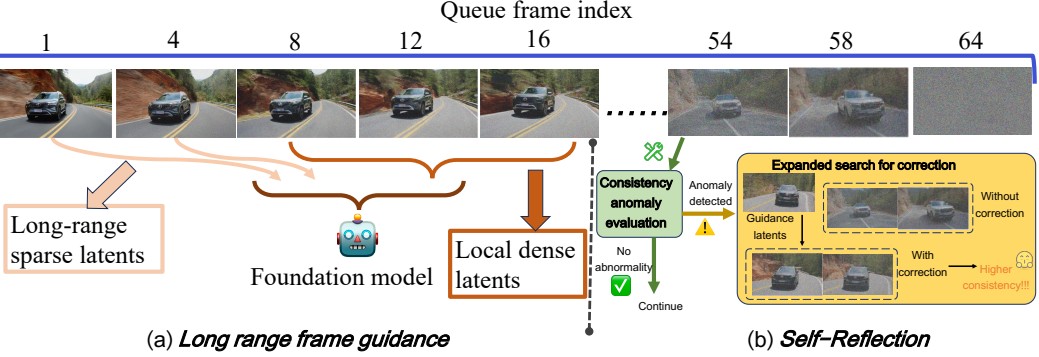

Queue frame index

(a) *Long range frame guidance*  (b) *Self-Reflection*

Figure A1: **Framework of our Dual Consistency Enhancement (DCE) mechanism, which consists of Long-Range Frame Guidance (a) and Self-Reflection (b) approaches. (a)** Long-Range Frame Guidance enables the foundation model to process long-range sparse latents and local dense latents simultaneously as input when handling local segments in a sliding manner. **(b)** The Self-Reflection approach starts from high-noise latents at the end of the queue and first performs consistency anomaly evaluation. When anomalies are detected (*e.g.*, the car's color changes to white whereas it was black in previous frames), expanded search for correction is conducted on these tail latents. After correction, the updated latents achieve higher consistency.

### A.1.3 DUAL CONSISTENCY ENHANCEMENT MECHANISM.

The modeling motivation and core concept of our Dual Consistency Enhancement (DCE) mechanisms are introduced in Sec. 3.3. In this section, we present the implementation details in the form of

---

**Algorithm 4** Initial latents construction in MIGA

---

**Require:** Denoising step count $T$, initial latents count $f_0$, zigzag width $L_{\text{zig}}$, denoising network $\epsilon_\theta(\cdot)$, and sampler $\Phi(\cdot)$

1: **Input**: initial clean latents $q_{\text{clean}} = \{\mathbf{z}_{\tau_0}^1; \ldots; \mathbf{z}_{\tau_0 0}^f\}$, time steps $t_{\text{clean}} = \underbrace{\{\tau_0; \ldots; \tau_0\}}_{f_0}$

2: **Output**: $\mathcal{Q}_{\text{init}} = \{\underbrace{\mathbf{z}_{\tau_e}^1, \cdots, \mathbf{z}_{\tau_e}^{L_{\text{zig}}}}_{L_{\text{zig}}}, \underbrace{\mathbf{z}_{\tau_{e+1}}^{L_{\text{zig}}+1}, \cdots, \mathbf{z}_{\tau_{e+1}}^{2L_{\text{zig}}}}_{L_{\text{zig}}}, \cdots, \underbrace{\mathbf{z}_{\tau_T}^{L-L_{\text{zig}}+1}, \cdots, \mathbf{z}_{\tau_T}^{L}}_{L_{\text{zig}}}\}$,

3: $\qquad\qquad t_{\text{init}} = \{\underbrace{\tau_e, \cdots, \tau_e}_{L_{\text{zig}}}, \underbrace{\tau_{e+1}, \cdots, \tau_{e+1}}_{L_{\text{zig}}}, \cdots, \underbrace{\tau_T, \cdots, \tau_T}_{L_{\text{zig}}}\}$

4: $\mathcal{Q}_{\text{init}} = \{\}, \quad t_{\text{init}} = \{\}$

5: # Manually add noise to the initial latents.

6: $n_{\text{zig}} = \lceil f_0/L_{\text{zig}} \rceil$

7: **for** $i = 1$ to $f_0$ **do**

8: $\qquad t_{\text{index}} = T - n_{\text{zig}} + \lceil (i-1)/L_{\text{zig}} \rceil$

9: $\qquad \mathbf{z}_{t_{\text{index}}}^i \leftarrow \Phi.\text{add\_noise}(q_{\text{clean}}[i], t_{\text{clean}}[i], \tau_{t_{\text{index}}})$

10: $\qquad \mathcal{Q}_{\text{init}}.\text{enqueue}(\mathbf{z}_{\tau_{t_{\text{index}}}}^i)$

11: $\qquad t_{\text{init}}.\text{enqueue}(\tau_{t_{\text{index}}})$

12: **end for**

13: # Progressively guide the generation of subsequent latents using existing latents.

14: **for** $i = 1$ to $T - n_{\text{zig}} - e$ **do**

15: $\qquad$ **for** $j = 1$ to $L_{\text{zig}}$ **do**

16: $\qquad\qquad \mathbf{z}_{\tau_T}^T \sim \mathcal{N}(0, \mathbf{I})$

17: $\qquad\qquad \mathcal{Q}_{\text{init}}.\text{enqueue}(\mathbf{z}_{\tau_T}^T)$

18: $\qquad\qquad t_{\text{init}}.\text{enqueue}(\tau_{t_T})$

19: $\qquad$ **end for**

20: $\qquad \mathcal{Q}_{\text{init}} \leftarrow \Phi(\mathcal{Q}_{\text{init}}, t_{\text{init}}; \epsilon_\theta)$

21: $\qquad$ **for** $j = 1$ to $\text{len}(t_{\text{init}})$ **do**

22: $\qquad\qquad t_{\text{init}}[j] = t_{\text{init}}[j] - 1$

23: $\qquad$ **end for**

24: **end for**

25: **return** $\mathcal{Q}_{\text{init}}, \quad t_{\text{init}}$

---

---

**Algorithm 5** Two-stage training-inference alignment mechanism in MIGA

---

**Require:** Denoising step count $T$, generation latents count $N$, zigzag width $L_{\text{zig}}$, denoising network $\epsilon_\theta(\cdot)$, and sampler $\Phi(\cdot)$

1: **Input:** $\mathcal{Q} = \{\underbrace{\mathbf{z}^1_{\tau_e}, \cdots, \mathbf{z}^{L_{\text{zig}}}_{\tau_e}}_{L_{\text{zig}}}, \underbrace{\mathbf{z}^{L_{\text{zig}}+1}_{\tau_{e+1}}, \cdots, \mathbf{z}^{2L_{\text{zig}}}_{\tau_{e+1}}}_{L_{\text{zig}}}, \cdots, \underbrace{\mathbf{z}^{L-L_{\text{zig}}+1}_{\tau_T}, \cdots, \mathbf{z}^{L}_{\tau_T}}_{L_{\text{zig}}}\}$,

2: $\qquad\qquad t = \{\underbrace{\tau_e, \cdots, \tau_e}_{L_{\text{zig}}}, \underbrace{\tau_{e+1}, \cdots, \tau_{e+1}}_{L_{\text{zig}}}, \cdots, \underbrace{\tau_T, \cdots, \tau_T}_{L_{\text{zig}}}\}$

3: **Output:** $\mathcal{Q}_{\text{gen}} = \{\mathbf{z}^1_{\tau_0}; \ldots; \mathbf{z}^N_{\tau_0}\}$

4: $\mathcal{Q}_{\text{gen}} = \{\}$

5: # Stage 1: Zigzag iterative denoising.

6: $n_{\text{iter}} = \lceil N/L_{\text{zig}} \rceil$

7: **for** $i = 1$ to $n_{\text{iter}}$ **do**

8: $\qquad Q \leftarrow \Phi(Q, t; \epsilon_\theta)$

9: $\qquad$ **for** $j = 1$ to $L_{\text{zig}}$ **do**

10: $\qquad\qquad \mathbf{z}^i_{\tau_0} \leftarrow \mathcal{Q}.\text{dequeue}()$

11: $\qquad\qquad \mathcal{Q}_{\text{gen}}.\text{enqueue}(\mathbf{z}^i_{\tau_0})$

12: $\qquad\qquad \mathbf{z}^T_{\tau_T} \sim \mathcal{N}(0, \mathbf{I})$

13: $\qquad\qquad \mathcal{Q}.\text{enqueue}(\mathbf{z}^T_{\tau_T})$

14: $\qquad$ **end for**

15: **end for**

16: # Stage 2: Denoising at a unified noise level.

17: $t = \{\}$

18: **for** $i = 1$ to $\text{len}(\mathcal{Q}_{\text{gen}})$ **do**

19: $\qquad t.\text{enqueue}(\tau_{e-1})$

20: **end for**

21: **for** $i = 1$ to $e - 1$ **do**

22: $\qquad \mathcal{Q}_{\text{gen}} \leftarrow \Phi(\mathcal{Q}_{\text{gen}}, t; \epsilon_\theta)$

23: $\qquad$ **for** $j = 1$ to $\text{len}(t)$ **do**

24: $\qquad\qquad t[j] = t[j] - 1$

25: $\qquad$ **end for**

26: **end for**

27: **return** $\mathcal{Q}_{\text{init}}, \quad t_{\text{init}}$

---

pseudocode. The pseudocode implementation can be understood with reference to the framework diagram shown in Fig. A1.

First, our Self-Reflection approach focuses on early high-noise latents along the queue dimension, performing consistency evaluation to promptly correct the latent anomalies. In conjunction with the previously described TTA iterative generation process, Self-Reflection is applied before each queue inference (*i.e.*, $Q \leftarrow \Phi(Q, t; \epsilon_\theta)$, see line 8 in Alg. 5). As shown in Alg. 6, we provide a detailed description of the implementation of the Self-Reflection approach. Note that the implementation of the Self-Reflection approach in Sec. 3.3 assumes $L_{\text{zig}} = 1$. A more general version is provided in Alg. 6.

Furthermore, the Long-Range Frame Guidance aproach targets low-noise latents at the queue head to facilitate interactions among distant latents, thereby improving video consistency. It is integrated into each queue inference step within the TTA iterative generation process (*i.e.*, $Q \leftarrow \Phi(Q, t; \epsilon_\theta)$, see line 8 in Alg. 5), and its implementation is detailed in Alg. 7.

### A.2 MULTI-PROMPT CONDITIONAL GENERATION

As discussed in Sec. 4.1, the flexibility of the frame-level autoregressive framework enables multi-text control by providing different text conditions to the latents at various temporal positions. Specifically, when performing inference over the queue, the expression involving the text conditioning $c$ is given by (for clarity, $c$ is omitted in Eq. 2):

$$\{\mathbf{z}_{\tau_0}^1, \ldots, \mathbf{z}_{\tau_{T-1}}^T\} = \Phi(\{\mathbf{z}_{\tau_1}^1, \ldots, \mathbf{z}_{\tau_T}^T\}, \{\tau_1, \ldots, \tau_T\}, c; \varepsilon_\theta). \quad \text{(A2)}$$

When there is only a single text condition, $c$ is a constant (*i.e.*, the feature of a single text prompt). However, with multiple text conditions, as the foundation model iterates over the queue with a sliding window, latents at different positions are guided by different $c$. For the case of $n_{\text{prom}}$ text prompts, *i.e.*, $c = \{c_i\}_{i=1}^{n_{\text{prom}}}$, each prompt sequentially controls the generation of $N_{\text{prom}}$ frames (resulting in $N = n_{\text{prom}} N_{\text{prom}}$ clean latents for the final video). Formally, suppose the foundation model is at position $l$ in the queue (*i.e.*, $s_{\text{index}}$ in Alg. 3 and Alg. 7), and the number of dequeued clean latents is $n_{\text{deq}}$. Then, the input text condition for the model is:

$$c_{\text{in}} = c\left[\left\lceil \frac{l + n_{\text{deq}}}{N_{\text{prom}}} \right\rceil\right]. \quad \text{(A3)}$$

As shown in Fig. A2, we present a case generated by our Wan2.1-based MIGA that contains three prompts.

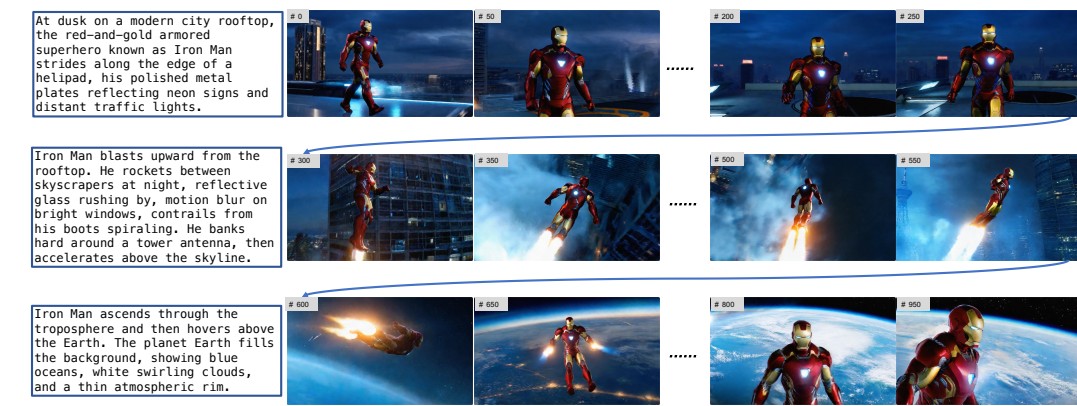

Figure A2: **Illustration of a multi-prompt controlled generation case produced by our Wan2.1-based MIGA.**

### A.3 IMPLEMENTATION ON DIFFERENT FOUNDATION MODELS

**VideoCrafter2-Based MIGA.** As an early and representative text-to-video foundation model, VideoCrafter2 (Chen et al., 2023; 2024b) is widely used as the backbone for existing train-free

---

**Algorithm 6** Self-reflection approach in MIGA

---

**Require:** Judgment index $f_{\mathrm{judg}}$, reference latents count $f_{\mathrm{ref}}$, evaluation latents count $f_{\mathrm{eval}}$, adjustment threshold $\delta_{\mathrm{adju}}$, guidance latents count $f_{\mathrm{guid}}$, zigzag width $L_{\mathrm{zig}}$, consistency score from the previous step $C_{\mathrm{score}}^0$, denoising network $\epsilon_\theta(\cdot)$, and sampler $\Phi(\cdot)$

1: **Input**: $\mathcal{Q} = \{\mathbf{z}^1; \ldots; \mathbf{z}^L\}$, $t = \underbrace{\{\tau^1; \ldots; \tau^L\}}_{L}$ # Without loss of generality, we do not distinguish the noise step of each latent.

2: **Output**: $\mathcal{Q}_{\mathrm{adju}} = \{\mathbf{z}^1; \ldots; \mathbf{z}^L\}, C_{\mathrm{score}}^0$

3: $\mathcal{Q}_{\mathrm{adju}} = \{\}$

4: # Evaluation

5: $q_{\mathrm{ref}} = \mathcal{Q}[f_{\mathrm{judg}} - f_{\mathrm{ref}} : f_{\mathrm{judg}} - 1]$

6: $q_{\mathrm{eval}} = \mathcal{Q}[f_{\mathrm{judg}} : f_{\mathrm{judg}} + f_{\mathrm{eval}} - 1]$

7: $C_{\mathrm{score}} = \mathrm{cosine\_similarity}(q_{\mathrm{ref}}, q_{\mathrm{eval}})$ # See Eq. 5 and Eq. 6.

8: **if** $C_{\mathrm{score}}^0 - C_{\mathrm{score}} \le \delta_{\mathrm{adju}}$ **then**

9:      # Do not perform extended sampling.

10:      $\mathcal{Q}_{\mathrm{adju}} = \mathcal{Q}$

11:      $C_{\mathrm{score}}^0 = C_{\mathrm{score}}$

12: **else**

13:      # Perform extended sampling.

14:      $q_{\mathrm{guid}} = \mathcal{Q}[f_{\mathrm{judg}} - f_{\mathrm{guid}} : f_{\mathrm{guid}} - 1]$

15:      $t_{\mathrm{guid}} = t[f_{\mathrm{judg}} - f_{\mathrm{guid}} : f_{\mathrm{guid}} - 1]$

16:      $\mathcal{Q}_{\mathrm{sample}} = \{\}$

17:      **for** $k = 1$ to $n_{\mathrm{samp}}$ **do**

18:          $\mathcal{Q}_{\mathrm{temp}} = q_{\mathrm{guid}}$

19:          $t_{\mathrm{temp}} = t_{\mathrm{guid}}$

20:          $n_{\mathrm{iter}} = \lceil (L - f_{\mathrm{judg}} + 1)/L_{\mathrm{zig}} \rceil$

21:          **for** $i = 1$ to $n_{\mathrm{iter}}$ **do**

22:              $\mathcal{Q}_{\mathrm{temp}} \leftarrow \Phi(\mathcal{Q}_{\mathrm{temp}}, t_{\mathrm{temp}}; \epsilon_\theta)$

23:              **for** $j = 1$ to $L_{\mathrm{zig}}$ **do**

24:                  $\mathbf{z}_{\tau_T}^T \sim \mathcal{N}(0, \mathbf{I})$

25:                  $\mathcal{Q}_{\mathrm{temp}}.\mathrm{enqueue}(\mathbf{z}_{\tau_T}^T)$

26:                  $t_{\mathrm{temp}}.\mathrm{enqueue}(\tau_T)$

27:              **end for**

28:          **end for**

29:          $\mathcal{Q}_{\mathrm{sample}}.\mathrm{enqueue}(\mathcal{Q}_{\mathrm{temp}})$

30:      **end for**

31:      # Determine whether correction is needed

32:      $C_{\mathrm{samp}} = \{\}$

33:      **for** $k = 1$ to $n_{\mathrm{samp}}$ **do**

34:          $q_{\mathrm{ref}} = \mathcal{Q}[f_{\mathrm{guid}} - f_{\mathrm{ref}} : f_{\mathrm{judg}} - 1]$

35:          $q_{\mathrm{eval}} = \mathcal{Q}[f_{\mathrm{guid}} : f_{\mathrm{guid}} + f_{\mathrm{eval}} - 1]$

36:          $C^k = \mathrm{cosine\_similarity}(q_{\mathrm{ref}}, q_{\mathrm{eval}})$

37:          $C_{\mathrm{samp}}.\mathrm{enqueue}(C^k)$

38:      **end for**

39:      $\mathrm{max}_{\mathrm{index}}, C^{\mathrm{max}} = \mathrm{max}(C_{\mathrm{samp}})$

40:      **if** $C^{\mathrm{max}} > C_{\mathrm{score}}$ **then**

41:          # Perform correction.

42:          $\mathcal{Q}[f_{\mathrm{judg}} :] = \mathcal{Q}_{\mathrm{sample}}[\mathrm{max}_{\mathrm{index}}][f_{\mathrm{ref}} :]$

43:          $C_{\mathrm{score}}^0 = C^{\mathrm{max}}$

44:      **else**

45:          # Do not perform correction.

46:          $\mathcal{Q}_{\mathrm{adju}} = \mathcal{Q}$

47:          $C_{\mathrm{score}}^0 = C_{\mathrm{score}}$

48:      **end if**

49:

50: **end if**

51: **return** $\mathcal{Q}_{\mathrm{adju}}, \quad C_{\mathrm{score}}^0$

---

---

**Algorithm 7** One-step inference over the latents queue incorporates our long-range frame guidance

---

**Require:** Denoising network $\epsilon_\theta(\cdot)$, initial latents count $f_0$, sliding window stride $l_{\text{stride}}$, sliding window stride $l_{\text{stride}}$, zigzag width $L_{\text{zig}}$, long-range guidance latents count $m_{\text{guid}}$ , and initial sampler $\phi(\cdot)$

1: **Input**: $\mathcal{Q} = \{\mathbf{z}^1; \ldots; \mathbf{z}^L\}$, $t = \underbrace{\{\tau^1; \ldots; \tau^L\}}_{L}$ # Without loss of generality, we do not distinguish the noise step of each latent.

2: **Output**: $\mathcal{Q}_{\text{gen}} = \{\mathbf{z'}^1; \ldots; \mathbf{z'}^L\}$ # $\mathbf{z'}^i$ represents $\mathbf{z}^i$ after one denoising step.

3: # Queue length equals denoising steps $T$.

4: $l_{\text{stride}} = [f_0/2]$

5: $n_{\text{iter}} = \lceil (T - f_0 + m_{\text{guid}})/l_{\text{stride}} \rceil + 1$

6: **for** $i = 1$ to $n_{\text{iter}}$ **do**

7:     **if** $i < n_{\text{iter}}$ **then**

8:         $s_{\text{index}} = (i - 1) \times l_{\text{stride}}$

9:         **if** $s_{\text{index}} <= n_{\text{iter}}$ **then**

10:             $e_{\text{index}} = s_{\text{index}} + f_0$

11:             $\mathcal{Q}_{\text{temp}} \leftarrow \phi(\mathcal{Q}[s_{\text{index}} : e_{\text{index}}], t[s_{\text{index}} : e_{\text{index}}]; \epsilon_\theta)$

12:             **for** $j = 1$ to $l_{\text{stride}}$ **do**

13:                 $\mathcal{Q}[s_{\text{index}} + j] = \mathcal{Q}_{\text{temp}}[j]$

14:             **end for**

15:         **else**

16:             $s_{\text{range}} = \min(m_{\text{guid}} L_{\text{zig}}, s_{\text{index}} - 1)$

17:             $s^0_{\text{list}} = \text{uniform\_sample}(\text{range}(s_{\text{index}} - s_{\text{range}}, s_{\text{index}} - 1), m_{\text{guid}})$ # Uniformly sample $m_{\text{guid}}$ position indices from the interval $(s_{\text{index}} - s_{\text{range}}, \ s_{\text{index}} - 1)$.

18:             $\mathcal{Q}_{\text{input}} = \{\}$

19:             $t_{\text{input}} = \{\}$

20:             **for** $j = 1$ to $m_{\text{guid}}$ **do**

21:                 $\mathcal{Q}_{\text{input}}.\text{enqueue}(\mathcal{Q}[s^0_{\text{list}}[j]])$

22:                 $t_{\text{input}}.\text{enqueue}(t[s^0_{\text{list}}[j]])$

23:             **end for**

24:             $e_{\text{index}} = s_{\text{index}} + f_0 - m_{\text{guid}}$

25:             $\mathcal{Q}_{\text{input}} = \text{concat}([\mathcal{Q}_{\text{input}}; \mathcal{Q}[s_{\text{index}} : e_{\text{index}}]])$

26:             $t_{\text{input}} = \text{concat}([t_{\text{input}}; t[s_{\text{index}} : e_{\text{index}}]])$

27:             $\mathcal{Q}_{\text{temp}} \leftarrow \phi(\mathcal{Q}[s_{\text{index}} : e_{\text{index}}], t[s_{\text{index}} : e_{\text{index}}]; \epsilon_\theta)$

28:             **for** $j = 1$ to $l_{\text{stride}}$ **do**

29:                 $\mathcal{Q}[s_{\text{index}} + j] = \mathcal{Q}_{\text{temp}}[m_{\text{guid}} + j]$

30:             **end for**

31:         **end if**

32:     **else**

33:         $s_{\text{index}} = T - f_0 + m_{\text{guid}}$

34:         $s_{\text{range}} = \min(m_{\text{guid}} L_{\text{zig}}, s_{\text{index}} - 1)$

35:         $s^0_{\text{list}} = \text{uniform\_sample}(\text{range}(s_{\text{index}} - s_{\text{range}}, s_{\text{index}} - 1), m_{\text{guid}})$

36:         $\mathcal{Q}_{\text{input}} = \{\}$

37:         $t_{\text{input}} = \{\}$

38:         **for** $j = 1$ to $m_{\text{guid}}$ **do**

39:             $\mathcal{Q}_{\text{input}}.\text{enqueue}(\mathcal{Q}[s^0_{\text{list}}[j]])$

40:             $t_{\text{input}}.\text{enqueue}(t[s^0_{\text{list}}[j]])$

41:         **end for**

42:         $e_{\text{index}} = s_{\text{index}} + f_0 - m_{\text{guid}}$

43:         $\mathcal{Q}_{\text{input}} = \text{concat}([\mathcal{Q}_{\text{input}}; \mathcal{Q}[s_{\text{index}} : e_{\text{index}}]])$

44:         $t_{\text{input}} = \text{concat}([t_{\text{input}}; t[s_{\text{index}} : e_{\text{index}}]])$

45:         $\mathcal{Q}_{\text{temp}} \leftarrow \phi(\mathcal{Q}[s_{\text{index}} : e_{\text{index}}], t[s_{\text{index}} : e_{\text{index}}]; \epsilon_\theta)$

46:         **for** $j = 1$ to $l_{\text{stride}}$ **do**

47:             $\mathcal{Q}[s_{\text{index}} + j] = \mathcal{Q}_{\text{temp}}[m_{\text{guid}} + j]$

48:         **end for**

49:     **end if**

50: **end for**

51: $\mathcal{Q}_{\text{gen}} = \mathcal{Q}$

52: **return** $\mathcal{Q}_{\text{gen}}$

---

long video generation methods (Qiu et al., 2023; Tan et al., 2025; Lu et al., 2024; Kim et al., 2024; Yang et al., 2025). Following its original denoising inference code, our main modification to equip it with frame-level autoregressive generation is to adjust its noise prediction model $\epsilon_\theta(\cdot)$ and sampler (*i.e.*, DDIM (Song et al., 2020)) $\phi(\cdot)$ in the noise prediction and denoising process. Specifically, the original $\epsilon_\theta(\cdot)$ receives latents with identical noise levels each time, and $\phi(\cdot)$ applies the same denoising operation to all latents. Our key change is that $\epsilon_\theta(\cdot)$ now receives latents with different noise levels in each inference step, where the noise level is determined by the frame index. During noise prediction, each frame's latents interact with their respective noise level conditions. In the denoising stage, $\phi(\cdot)$ also processes latents for different frames separately, conditioning on their noise levels.

**Wan2.1-Based MIGA.** With the continuous evolution of text-to-video foundation models, train-free long video generation frameworks should also be adapted to these newer models. To this end, we migrate MIGA to the latest available model, Wan2.1 (Wan et al., 2025). Similar to the VideoCrafter2-based MIGA, the core modifications involve adjusting the noise prediction model $\epsilon_\theta(\cdot)$ and the sampler $\phi(\cdot)$ in the noise prediction and denoising process. The key difference is that Wan2.1 employs UniPC (Zhao et al., 2023) as its default sampler, which requires higher-order computations. Consequently, both $\epsilon_\theta(\cdot)$ and $\phi(\cdot)$ need to store and utilize information from previous steps during each operation.

**Discussion on the Generalizability of Our Method.** The frame-level autoregressive generation framework we adopt inherently requires models to handle latents with noise levels varying across frames. Our MIGA can be migrated to VideoCrafter2 and Wan2.1 because both their noise conditions and text conditions interact with latents via cross-attention. Specifically, latents can incorporate noise timestep conditions by distinguishing frame indices, while all latents are treated as a whole for text conditioning. However, We observe that this frame-level autoregressive generation framework is difficult to apply to certain foundation models (Kong et al., 2024; Yang et al., 2024) based on the MMDiT architecture (Esser et al., 2024). The main reason is that these models concatenate text and video features, and jointly interact with the noise timestep condition. To guide latents of different frames with distinct noise levels, it is necessary to introduce noise conditions with varying timesteps. However, since text features cannot be distinguished at the frame level, this noise information cannot effectively interact with the text features.. Fig. A3 illustrates our approach of injecting an intermediate timestep into the text features to enable the migration of the frame-level autoregressive generation framework to CogVideoX-5B (Yang et al., 2024), which is based on the MMDiT architecture. As shown, this results in abnormal video outputs.

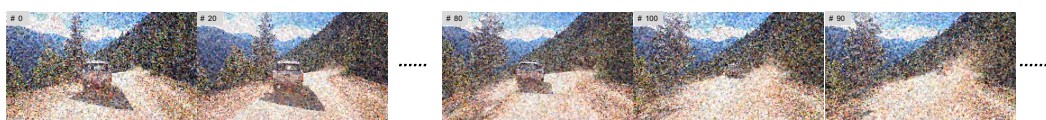

An SUV traversing serpentine mountain roads.

Figure A3: **Illustration of a bad case resulting from migrating frame-level autoregressive generation to CogVideoX, a model based on the MMDiT architecture.**

# B FURTHER DETAILS ON EXPERIMENTAL ANALYSIS

## B.1 IMPLEMENTATION DETAILS OF ABLATION STUDIES

In Sec. 4.3, we conduct comprehensive ablation studies on the VideoCrafter2-based MIGA using VBench to evaluate the effectiveness of our proposed mechanism design. In this subsection, we provide detailed implementation information for each setting.

**Study on Core Mechanism Designs.** The core contribution of our work lies in the introduction of the novel Two-Stage Training-Inference Alignment (TTA) and Dual Consistency Enhancement (DCE) mechanisms. In Tab. 3, we use FIFO-Diffusion as the baseline setting without TTA or DCE, which serves as the baseline for our ablation study. For the TTA-only setting, we add Stage 1 zigzag iterative denoising with $L_{\text{zig}} = 4$ and Stage 2 unified noise level denoising with $\tau_e = 10$ to the baseline. For the DCE-only setting, we introduce a self-reflection method with $\delta_{\text{adju}} = 0.01$ and a long-range frame guidance method with $m_{\text{guid}} = 4$ on top of the baseline. Finally, combining both mechanisms yields our final method.

**Study on TTA.** Our TTA mechanism consists of two stages: stage 1 employs zigzag iterative denoising, and stage 2 applies denoising at a unified noise level. In Tab. 6, FIFO-Diffusion is used as the baseline, consistent with the setting in Tab. 3. For the "+Stage 1" setting, we add only stage 1 with $L_{\mathrm{zig}} = 4$ to the baseline. Building upon this, by incorporating stage 2 with $\tau_e = 10$, we obtain the "+Stage 2" setting.

Next, we conduct ablation studies on the two core hyperparameters in TTA, *i.e.*, $L_{\mathrm{zig}}$ and $\tau_e$ (where $\tau_e$ determines the number of steps in stage 2, $(e-1)$). Considering the high computational cost of long video generation, these experiments are carried out on a subset of the evaluation set, selected by randomly sampling 50% of the prompts from the full evaluation prompts. For the baseline, *i.e.*, $L_{\mathrm{zig}} = 1$, we report the performance of FIFO-Diffusion on this subset. Then, we progressively increase $L_{\mathrm{zig}}$ and present the results in Tab. 4. Fig. 5 illustrates the impact of varying the number of stage 2 denoising steps on model performance, highlighting the overall score metric across different settings. The detailed results for each individual metric under these settings are presented in Tab. A1. For the baseline setting ($e = 1$), we report the performance of FIFO-Diffusion on this evaluation subset. Then, we further analyze the results for $e = 5, 10, 15, 25, 30$ based on this baseline. When $e$ is increased to the total denoising steps (64), stage 1 is entirely omitted, and only stage 2 is performed. As illustrated in Alg. 8, $N$ random noises are first initialized, and then the queue undergoes 64 inference steps (each executed by sliding the window across the queue with the foundation model), ultimately producing $N$ clean latents in one pass. It is important to note that this approach completely loses the autoregressive property in generation, whereas stage 1 (when retained) preserves the autoregressive nature. Compared with performing only stage 2, stage 1—through its initialization and iterative generation (see Alg. 4 and Alg. 5)—enables the cleaner latents to implicitly guide the generation of subsequent high-noise latents, thereby enhancing the consistency of the generated videos. This connection among latents ensures video coherence, whereas only stage 2, by synchronously denoising $N$ independent noise latents, results in weak latent correlation and poorer consistency.

A qualitative explanation of the synergy between these two stages is as follows: stage 1 is responsible for the early denoising of latents, leveraging the autoregressive mechanism to build connections among latents and maintain semantic and spatial consistency (Qiu et al., 2023). Subsequently, stage 2 completes the final denoising process. Once the overall content has been established, stage 2 matches the input conditions seen during training (*i.e.*, the noise span is 1), which contributes to improved visual details in the generated videos. As shown in Fig. 4 (a–c), stage 2 effectively suppresses visual artifacts such as noise in the output videos.

Table A1: Ablation study on steps of stage 2.

| Steps | S.C. | B.C. | M.S. | T.F. | O.S. |
|---|---|---|---|---|---|
| 0 | 94.23 | 94.52 | 97.98 | 96.47 | 95.80 |
| 5 | 93.58 | 95.33 | 98.19 | 96.68 | 95.95 |
| 10 | 93.64 | 95.73 | 98.11 | 96.36 | 95.96 |
| 15 | 94.05 | 95.37 | 98.23 | 96.74 | 96.10 |
| 20 | 94.07 | 95.42 | 98.24 | 96.80 | 96.13 |
| 25 | 93.54 | 95.58 | 98.30 | 97.09 | 96.13 |
| 30 | 93.92 | 95.70 | 98.17 | 96.64 | 96.11 |
| 64 | 89.91 | 94.45 | 97.16 | 95.47 | 94.25 |

**Study on DCE.** Our DCE mechanism is composed of self-reflection and long-range frame guidance. Fig. 6 shows the impact of different adjustment thresholds $\delta_{\mathrm{adju}}$. The baseline (*i.e.*, $\delta_{\mathrm{adju}} = 0.07$, where no extended search is performed) is consistent with that used in the above TTA experiments. Based on this baseline, we progressively decrease $\delta_{\mathrm{adju}}$, which increases the frequency of extended search. Fig. 6 focuses on the overall score (O.S.), while detailed performance on each metric is provided in Tab. A2. Tab. 5 presents the effect of the number of long-range guiding frames $m_{\mathrm{guid}}$ on model performance. The same baseline as in Fig. 6 is adopted, and results under different settings are obtained by gradually increasing $m_{\mathrm{guid}}$. The computational costs introduced by these two approaches are discussed in Sec. B.3.

---

**Algorithm 8** Inference procedure with stage 2 only

---

**Require:** Denoising step count $T$, generation latents count $N$, denoising network $\epsilon_\theta(\cdot)$, and sampler $\Phi(\cdot)$

1: **Input**: $\mathcal{Q} = \{\mathbf{z}_{\tau_T}^1; \mathbf{z}_{\tau_T}^2; \ldots; \mathbf{z}_{\tau_T}^N\}$, $t = \underbrace{\{\tau_T; \tau_T; \ldots; \tau_T\}}_{N}$

2: **Output**: $\mathcal{Q}_{\text{gen}} = \{\mathbf{z}_{\tau_0}^1; \ldots; \mathbf{z}_{\tau_0}^N\}$

3: **for** $i = 1$ to $T$ **do**

4:     $Q \leftarrow \Phi(Q, t; \epsilon_\theta)$

5:     **for** $j = 1$ to $T$ **do**

6:         $t[j] = t[j] - 1$

7:     **end for**

8: **end for**

9: $\mathcal{Q}_{\text{gen}} = \mathcal{Q}$

10: **return** $\mathcal{Q}_{\text{gen}}$

---

Table A2: Ablation study on the threshold $\delta_{\text{adju}}$.

| Threshold | S.C. | B.C. | M.S. | T.F. | O.S. |
|---|---|---|---|---|---|
| 0.001 | 95.91 | 96.04 | 98.55 | 97.93 | 97.11 |
| 0.003 | 95.69 | 96.27 | 98.52 | 97.87 | 97.09 |
| 0.005 | 95.60 | 95.92 | 98.58 | 97.98 | 97.02 |
| 0.007 | 95.19 | 95.96 | 98.53 | 97.89 | 96.89 |
| 0.01 | 95.44 | 96.06 | 98.58 | 97.97 | 97.01 |
| 0.03 | 94.51 | 95.01 | 98.53 | 97.90 | 96.49 |
| 0.05 | 93.70 | 94.61 | 98.40 | 97.70 | 96.10 |
| 0.07 | 94.23 | 94.52 | 97.98 | 96.47 | 95.80 |

### B.2 Qualitative Comparison between VideoCrafter2-based MIGA and Wan2.1-based MIGA

In Sec. 4.2, we present the results of Wan-2.1-based MIGA and VideoCrafter2-based MIGA on VBench and NarrLV. As reported in Tab. 1, one noteworthy observation on the VBench results is that, contrary to intuition, MIGA built on the latest foundation model Wan-2.1 demonstrates weaker performance in subject and background consistency than MIGA based on the earlier foundation model VideoCrafter2. The main reason is that VideoCrafter2 primarily generates animation-style videos, where maintaining long-term consistency is relatively easier than in the realistic-style videos produced by Wan-2.1. To provide a clearer explanation, Fig. A4 showcases the generation results from both approaches for the same prompt, along with their VBench evaluation scores. As seen from the comparison between Fig. A4 (a) and (b), VideoCrafter2-based MIGA tends to produce more animation-like videos, while Wan-2.1-based MIGA generates content with richer texture details. Correspondingly, the former achieves higher scores in subject and background consistency, while the latter, benefiting from Wan-2.1's capability to generate coherent video content, attains better performance in motion smoothness and temporal flicker metrics. Moreover, the evaluation results on NarrLV further demonstrate Wan-2.1's strong ability to generate videos with richer narrative content.

### B.3 Computational Efficiency Analysis

FIFO-Diffusion (Kim et al., 2024) demonstrates the advantages of the frame-level autoregressive generation framework in terms of memory usage and inference time. Building on this, we analyze the computational efficiency of our method. Specifically, when only the Two-Stage Training-Inference Alignment (TTA) mechanism is introduced, the computational efficiency remains identical to that of the original FIFO-Diffusion. This is because, for each latent, the required number of denoising steps is $T$ in both approaches, resulting in equal computational cost for generating videos of the same length. As shown in Tab. 3, with comparable computational efficiency, our TTA mechanism yields a notable performance improvement (overall score increases by 2.03%), thereby demonstrating its effectiveness.

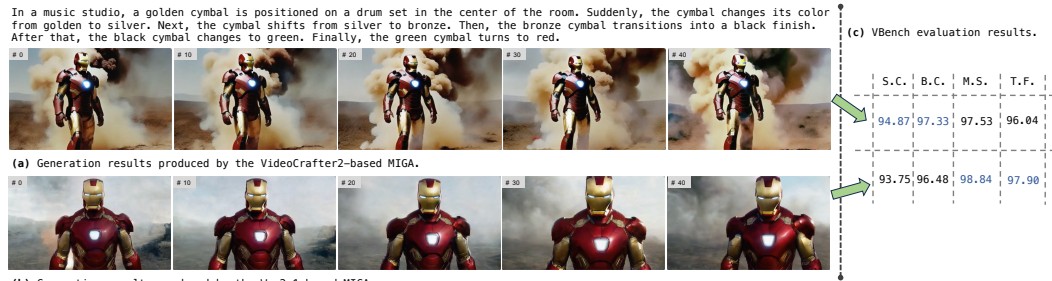

In a music studio, a golden cymbal is positioned on a drum set in the center of the room. Suddenly, the cymbal changes its color from golden to silver. Next, the cymbal shifts from silver to bronze. Then, the bronze cymbal transitions into a black finish. After that, the black cymbal changes to green. Finally, the green cymbal turns to red.

**(a)** Generation results produced by the VideoCrafter2-based MIGA.

**(b)** Generation results produced by the Wan2.1-based MIGA.

**(c)** VBench evaluation results.

| | S.C. | B.C. | M.S. | T.F. |
|---|---|---|---|---|
| | 94.87 | 97.33 | 97.53 | 96.04 |
| | 93.75 | 96.48 | 98.84 | 97.90 |

Figure A4: **Illustration of qualitative comparison between VideoCrafter2-based MIGA and Wan2.1-based MIGA (a–b), along with corresponding VBench evaluation results (c).** S.C., B.C., M.S., and T.F. denote subject consistency, background consistency, motion smoothness, and temporal flicker, respectively. The superior result for each metric is highlighted in blue.

Building upon TTA, our Dual Consistency Enhancement (DCE) mechanism introduces the additional computational cost to further improve the quality of generated videos. Specifically, the long-range frame guidance approach mainly affects the inference process for the maintained queue. Without this mechanism, as shown in Alg. 3, completing one queue inference requires the foundation model to perform $n_{\text{iter}} = \lceil (T - f_0)/l_{\text{stride}} \rceil + 1$ noise prediction steps, where $l_{\text{stride}} = [f_0/2]$. When long-range frame guidance is enabled, as shown in Alg. 7, the foundation model needs to perform $n_{\text{iter}} = \lceil (T - f_0 + m_{\text{guid}})/l_{\text{stride}} \rceil + 1$ noise prediction steps per queue inference. It can be seen that the additional number of noise prediction steps introduced by long-range frame guidance is small, *i.e.*, the extra computational overhead is negligible. Nevertheless, Tab. 5 demonstrates that long-range frame guidance yields considerable performance gains. The self-reflection approach belongs to the scope of Test-Time-Scaling (TTS) technologies, which aim to improve generation quality by increasing inference time. Here, each noise prediction by the model is treated as a basic computational unit. Specifically, the additional computational burden is mainly introduced through extended sampling. As shown in Alg. 6, in each generation process, the number of extended sampling iterations $n_{\text{adju}}$ is determined by the threshold $\delta_{\text{adju}}$. Once extended sampling is triggered, for each of the $n_{\text{samp}}$ samples, the foundation model must perform $\lceil (L - f_{\text{judg}} + 1)/L_{\text{zig}} \rceil$ noise prediction steps. Thus, the total number of extra noise predictions required by the self-reflection method is:

$$n_{\text{adju}} \times n_{\text{samp}} \times \left( \lceil (L - f_{\text{judg}} + 1)/L_{\text{zig}} \rceil \right). \tag{A4}$$

As an optional mechanism, the degree of TTS can be flexibly controlled by adjusting $n_{\text{adju}}$ (via modifying $\delta_{\text{adju}}$) and $n_{\text{samp}}$. As shown in Fig. 6, model performance improves as computational cost increases.

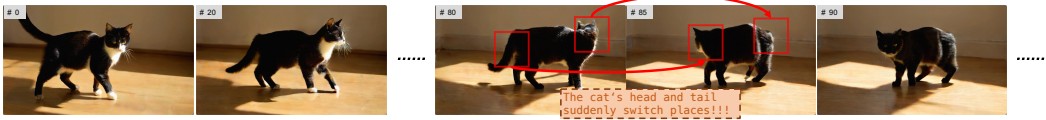

The cat's head and tail suddenly switch places!!!

A domestic cat walks slowly and gracefully across a sun-drenched wooden floor.

Figure A5: **Illustration of a bad case reflecting the limitations of our MIGA.**

### B.4 More Qualitative Results

Fig. 1 presents three long video cases generated by MIGA based on Wan2.1-1.3B. Additionally, more generation cases produced by MIGA using Wan2.1-1.3B and VideoCrafter2 are shown in Fig. A6 and Fig. A7. All these generated videos are approximately one minute in length. According to their default fps settings, Wan2.1-based MIGA produces long videos with 1000 frames, while VideoCrafter2-based MIGA generates long videos with 600 frames.

## C Limitations and Future Work

Our proposed MIGA provides an effective train-free approach for extending the generated length of existing foundation models. While longer video duration offers greater space for content creation, it

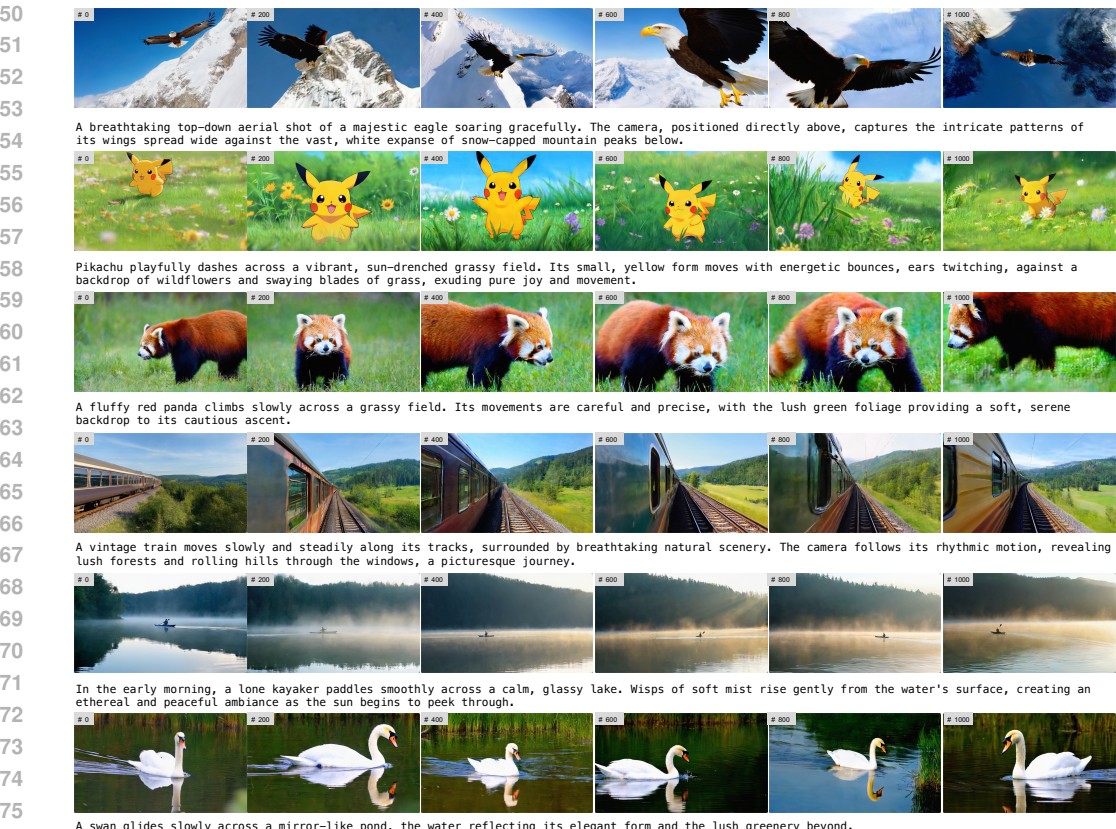

Figure A6: **Long video cases generated by Wan2.1-based MIGA.**

also increases the risk of unintended model behaviors. As shown in Fig. A5, the beginning of the generated video follows the text prompt well, with a cat walking from left to right. However, after some time, the cat's head and tail suddenly switch places. This phenomenon can be regarded as a hallucination (Chu et al., 2024; Bai et al., 2024) of the video generation model, or as evidence of the lack of underlying physical knowledge (Lin et al., 2025; Bansal et al., 2024). Such issues are not only specific to long video generation tasks but also represent a major challenge for the entire field of video generation (Kang et al., 2024). In future work, we aim to incorporate additional conditioning signals beyond text instructions to enable the generation of more realistic long videos.

## D USAGE OF LARGE LANGUAGE MODELS

In this paper, large language models (GPT-4o) are used solely for polishing the writing of our manuscript.

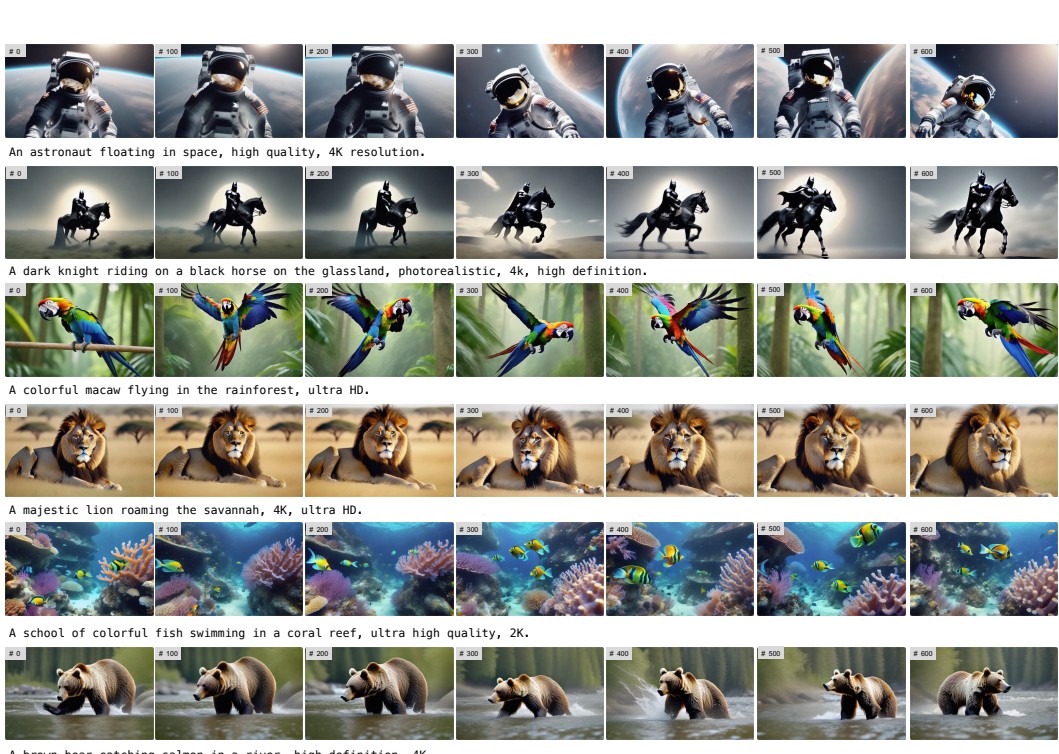

An astronaut floating in space, high quality, 4K resolution.

A dark knight riding on a black horse on the glassland, photorealistic, 4k, high definition.

A colorful macaw flying in the rainforest, ultra HD.

A majestic lion roaming the savannah, 4K, ultra HD.

A school of colorful fish swimming in a coral reef, ultra high quality, 2K.

A brown bear catching salmon in a river, high definition, 4K.

Figure A7: **Long video cases generated by VideoCrafter2-based MIGA.**

