# OpenReview forum: "MIGA: Make Train-Free Infinite Frame Generation Great Again for Consistent Long Videos"
_ICLR.cc/2026/Conference — Submitted to ICLR 2026_

### Official Review · Reviewer_qgSH · 2025-10-31

**Soundness:** 2
**Presentation:** 3
**Contribution:** 4
**Rating:** 4
**Confidence:** 4

**Summary:**

This paper presents MIGA, a training-free infinite-frame video generation method that extends foundation diffusion models for consistent long videos. It introduces two key designs: (1) a Two-Stage Training–Inference Alignment mechanism that reduces the noise-span mismatch between training and inference via zigzag denoising and unified refinement, and (2) a Dual Consistency Enhancement mechanism combining self-reflection–based anomaly correction and long-range frame guidance. Experiments on VBench and NarrLV benchmarks show that MIGA achieves state-of-the-art performance, improving subject and background consistency over prior training-free methods like FIFO-Diffusion and FreeLong, while maintaining infinite-frame generation capability.

**Strengths:**

1. The two-stage denoising process is intuitive yet effective, reducing the noise span mismatch with minimal modification to the inference pipeline.
2. The proposed self-reflection mechanism uses cosine similarity in latent space, eliminating dependence on external models like DINO and improving computational efficiency.
3. The figures and diagrams are highly informative and well-designed.

**Weaknesses:**

1. The proposed framework lacks a clear overarching motivation. Its components (e.g., TTA, self-reflection, and long-range frame guidance) appear to be designed independently rather than forming a unified, coherent approach to long video generation.
2. The paper does not demonstrate strong scientific contributions; the proposed method resembles more of an engineering pipeline than a conceptually novel framework.
3. Given the complexity of the pipeline, the authors should include a comparison of computational or time complexity with other sota methods.

minor weakness:
1. Line 161 mentions "its length L equals the total number," but L seems absent from the context; did you mean T?

**Questions:**

1. What is the key contribution of the paper, and which component highlights it?
2. How do the proposed methods relate and work together to address the research problem, specifically the train-inference gap?

---

> ### Author Response · Authors · 2025-11-21
> **Response to Reviewer qgSH (Part 1 of 2)**
>
> **Dear Reviewer qgSH,**
>
> Thank you for taking the time to review our work. We appreciate your recognition of the intuitive and effective nature of our Two-Stage Training–Inference Alignment (TTA) mechanism, the computational efficiency of our Dual Consistency Enhancement (DCE) mechanism, our state-of-the-art performance, and the informative, well-designed figures and diagrams.
> Below, we will provide detailed responses to your remaining concerns.
>
> ---
>
> ### **Q1: Analysis of Framework Unification**
>
> Thank you for your constructive feedback regarding the overall unification of our framework. We would like to further clarify the integrated design of MIGA within the context of the adopted frame-level autoregressive generation paradigm.
>
> 1. **Frame-level Autoregressive Framework:** Figure 2 (a) of our manuscript illustrates the autoregressive generation process of this framework.
>    - At the core of this process is the maintenance of a structured queue of noisy frames, with noise intensity increasing along the temporal axis. At each inference step, the noise levels of all frames are simultaneously reduced, allowing the clean frame to be popped from the queue head. By repeating this procedure iteratively, frame-level autoregressive generation is achieved.
>    - Furthermore, during each inference operation on the queue, since foundation models can only process a limited number of frames at a time, a sliding window is employed to sequentially refine all noisy frames in the queue.
>
> 2. **Unification of MIGA Modules:** Corresponding to the overall queue-level inference process and the local frame-level sliding-window denoising within each queue, our core TTA and DCE mechanisms are specifically designed to optimize these two aspects.
>    - The TTA mechanism divides the queue-level inference into two stages, each characterized by different noise intensity distributions across the frames. Through coordinated operation, TTA reduces the noise span fed to the model at each step, effectively transferring the short-term generative capabilities of foundation models to the long-video generation scenario.
>    - The DCE mechanism focuses on denoising within each queue iteration by employing long-range guidance from low-noise frames and timely self-reflection correction for high-noise early frames. This mechanism effectively enhances consistency in long video generation.
>
> As depicted in Figure 2(b), TTA primarily focuses on the vertical dimension (queue-level inference), while DCE operates along the horizontal dimension (frame-level denoising within each queue; see also Figure A.1 for a detailed illustration).
> Together, these two mechanisms form a unified and coherent generative framework that enables the consistent and high-quality generation of long videos.
>
> ---
>
> ### **Q2: Key Contributions and Novelty of MIGA**
>
> Thanks for your attention to the novelty and contributions of our approach. Building on our explanations in Q1 regarding the frame-level autoregressive framework and the unification of MIGA modules, we provide the following additional clarifications:
>
> 1. **Optimization Approaches within the Frame-level Autoregressive Framework:** As discussed in Q1, the training-free frame-level autoregressive framework achieves long video generation primarily through specific designs applied to the inference process, without requiring changes to the model architecture. We speculate that this emphasis on inference design may give the impression of an engineering pipeline. It is important to note that subsequent works following this framework (e.g., ScalingNoise [1]) have similarly focused on improving inference procedures, yet such optimization approaches have consistently yielded meaningful performance gains.
>
> 2. **Our Key Contributions and Novelty:** Our primary contributions lie in the systematic analysis of the challenges in bridging the training–inference gap and maintaining long-term consistency, which are well-known and difficult problems for practical deployment, yet remain insufficiently addressed by previous approaches (as noted by Reviewer cRYb).  To address these issues, we introduce the TTA mechanism to proactively reduce the training-inference gap by optimizing the noise span, and we design the DCE mechanism to enhance long-term consistency. Furthermore, comprehensive performance comparisons (Section 4.2) and ablation studies (Section 4.3) demonstrate the effectiveness of our approach. We contend that our focus on these core challenges, together with the introduction of effective and practical solutions, substantiates the novelty and significance of our work.

---

> > ### Author Response · Authors · 2025-11-21
> > **Response to Reviewer qgSH (Part 2 of 2)**
> >
> > ### **Q3: Computational Efficiency Analysis**
> >
> > We appreciate your constructive suggestion to conduct a computational efficiency analysis. First, MIGA follows the frame-level autoregressive framework represented by FIFO-Diffusion [2]. Compared to models that support only limited-length video generation, FIFO-Diffusion has demonstrated advantages in memory consumption and inference speed.
> >
> > Based on this, we focus on comparing MIGA and FIFO-Diffusion in terms of computational efficiency (average time to generate one frame, $M_t$) and performance (Overall Score, O.S., on VBench). The results are presented in the table below, where "#2" denotes MIGA without the DCE mechanism. Since the Self-Reflection method within the DCE mechanism serves as a test-time scaling technique, it intentionally increases computational overhead to achieve higher performance and is therefore analyzed separately.
> >
> > The results show that, without the DCE mechanism, MIGA achieves similar computational efficiency to FIFO-Diffusion (+0.05s), while substantially improving performance (+1.73). Furthermore, the addition of DCE further improves performance at the cost of increased computational overhead. These results demonstrate that MIGA provides a favorable trade-off between performance and computational efficiency.
> >
> > | # | Setting |$M_t$  | O.S.   |
> > |---|--------|--------|-------|
> > | 1 | FIFO-Diffusion     | 7.48s  | 95.02  |
> > | 2 | MIGA w/o DCE       | 7.53s  | 96.75  |
> > | 3 | MIGA               | 9.16s  | 97.82  |
> >
> > ---
> >
> > ### **Q4: Clarification of Symbol L**
> >
> > Thank you for your careful review. As stated in the manuscript, "its length L equals the total number of denoising steps T," which is the first appearance of L in the text. Here, L refers to the length of the queue.
> >
> > ---
> >
> > We sincerely hope that our responses have addressed your concerns, and we would appreciate it if you could reconsider your rating. If you have any further feedback or require additional clarification, please feel free to let us know.
> >
> >
> > ---
> >
> > [1]. Scalingnoise: Scaling inference-time search for generating infinite videos, Yang et al., arXiv 2503.16400.
> >
> > [2]. Fifo-diffusion: Generating infinite videos from text without training, Kim et al.,  NeurIPS 2024.

---

### Official Review · Reviewer_PNPU · 2025-10-31

**Soundness:** 3
**Presentation:** 3
**Contribution:** 3
**Rating:** 4
**Confidence:** 3

**Summary:**

The paper introduces MIGA, a train-free framework that turns short-video diffusion models into infinite-frame generators with constant memory. It addresses two key issues: the training–inference gap and long-term consistency. First, a Two-Stage Training-Inference Alignment (TTA) reduces the span of noise levels the model must handle at inference: Stage-1 maintains a zigzag latent queue so noise changes more slowly across frames; Stage-2 performs unified-level denoising once all latents reach the same noise level, closely matching the training condition. This yields fewer artifacts and drift and is detailed with an explicit queue-update algorithm.
Second, Dual Consistency Enhancement (DCE) combines Self-Reflection (detects anomalies on high-noise tail latents and selectively re-samples to correct them) and Long-Range Frame Guidance (feeds sparse, cleaner anchor frames alongside local windows to promote distant-frame interaction). DCE adds little overhead for guidance and offers tunable test-time scaling for self-reflection; ablations show consistent gains on VBench metrics (subject/background consistency, motion smoothness, etc.). Demonstrations include ~1000-frame videos on Wan2.1-1.3B and ~600-frame on VideoCrafter2

**Strengths:**

MIGA’s big win is that it’s truly train-free yet still turns short-video diffusion models into infinite-frame generators with constant memory. It closes the training–inference gap with a neat two-step trick— a zigzag latent queue to keep neighboring frames at similar noise levels, followed by unified-level denoising—so no drift or flicker as videos get longer. Then it boosts long-range consistency with Dual Consistency Enhancement: lightweight self-reflection to catch and fix anomalies on the fly, plus long-range frame guidance that pulls in clean anchor frames to keep characters and backgrounds stable. It’s plug-and-play (works with VideoCrafter2, Wan2.1, etc.), scales to 600–1000+ frames, and shows solid gains on VBench/Narrative metrics—practical, efficient, and easy to adopt.

**Weaknesses:**

1. Novelty vs. prior art: The “train-free long-video” direction already includes methods like FreeNoise / FreeLong / FreePCA (extend duration by re-using or transforming noise) and Diffusion-Forcing / AR-Diffusion / FIFO-Diffusion (queue/FIFO autoregression with constant memory). MIGA builds on this paradigm by adding alignment and consistency enhancements (TTA + DCE), so its contribution is more incremental than first-of-its-kind on “∞-frame with constant memory.”
2. Model diversity: Experiments are run on two base models only—VideoCrafter2 and Wan2.1-1.3B—with some discussion that stylistic differences can sway different metrics. Evidence of broad cross-model generalization is therefore limited.
3. Benchmark/data breadth: The evaluation focuses mainly on VBench (subject/background consistency, motion smoothness, flicker) and NarrLV (narrative metrics) against baselines like FIFO-Diffusion, FreeLong, FreePCA, and ScalingNoise. The benchmark surface is relatively narrow—there’s little coverage of diverse, real-world datasets or large-scale human studies—so external validity remains to be strengthened. The appendix also shows failure cases (e.g., structural mismatches on objects), indicating consistency is not fully solved.

**Questions:**

See Weakness

---

> ### Author Response · Authors · 2025-11-21
> **Response to Reviewer PNPU**
>
> **Dear Reviewer PNPU,**
>
> Thank you for your time and effort in reviewing our work. We appreciate your recognition of our core motivation to address the training–inference gap and long-term consistency challenges in train-free infinite-frame video generation. We are also grateful for your positive remarks on our neat Two-Stage Training-Inference Alignment (TTA) mechanism and Dual Consistency Enhancement (DCE), both of which are plug-and-play, practical, efficient, easy to adopt, and demonstrate solid gains in our experimental results. In response to your specific concerns, we have promptly conducted a thorough analysis and hope that our following responses will address them.
>
> ---
>
> ### **Q1: Novelty of MIGA Compared to Prior Work**
>
> Thank you for pointing out again the relevant prior works in the "train-free long-video" domain; these works were cited and discussed in our initial manuscript. Numerous studies in this area demonstrate that the task holds much promise, but still faces considerable challenges and unresolved questions. While methods such as FIFO-Diffusion pioneered the "infinite-frame with constant memory" paradigm, subsequent works (e.g., ScalingNoise [1])  under this paradigm have revealed several limitations within the framework.
>
> Our primary contributions lie in the systematic analysis of the challenges in maintaining long-term consistency and bridging the training–inference gap, which are critical for practical deployment yet remain insufficiently addressed by previous approaches. As Reviewer cRYb notes, preserving long-term consistency is a well-known and difficult problem. In response to these issues, we propose targeted improvements via our TTA and DCE mechanisms. In addition, comprehensive performance comparisons (Section 4.2) and ablation studies (Section 4.3) demonstrate the effectiveness of our approach. We contend that our focus on core challenges in the field, together with the introduction of effective and practical solutions, substantiates the novelty and significance of our work.
>
> ---
>
> ### **Q2: Analysis of Model Generalizability**
>
> Thanks for your attention to the generalizability of our approach. In line with established practices in the field, we offer the following additional clarifications:
>
> 1. As noted in your previous comment, prior related works such as FreeLong, FreePCA, and FIFO-Diffusion generally conduct experiments only on the VideoCrafter foundation model. Differently, our MIGA is applied to both VideoCrafter and the latest Wan2.1 model, thereby demonstrating superior generalizability compared to earlier approaches.
>
> 2. Furthermore, as discussed in Appendix A.3 of our initial manuscript, we have analyzed the generalizability of this frame-level autoregressive architecture. Specifically, due to inherent characteristics of such architectures, they are currently not compatible with models that concatenate text and video features for self-attention (see Appendix A.3 for specific reasons and details). For models employing cross-attention between text and video features, such architectures can be readily adapted.
>
> ---
>
> ### **Q3: Human Evaluation Experiments**
>
> Thank you for your constructive suggestions. To address your concerns regarding external validity, we conducted a large-scale user study. Specifically, we selected our MIGA and its improved baseline, FIFO-Diffusion, for comparative analysis. We randomly chose 48 evaluation prompts as generation conditions, resulting in 48 pairs of videos. Eight annotators were invited to compare each video pair across four dimensions: subject consistency, background consistency, motion smoothness, and temporal flicker. For each dimension, annotators identified the video with better performance, or indicated if both were equivalent.
>
> |Metric|MIGA is Better| Tie |FIFO_Diffusion is Better |
> |--|--|---|---|
> |Subject Consistency| 62.23%| 21.88%|15.89% |
> |Background Consistency| 61.72%|20.83%|17.45%|
> |Motion Smoothness| 66.14%| 19.79%|14.06%|
> |Temporal Flicker| 66.14% |17.70%|16.15%|
>
> As shown in the above table, our method demonstrates significant improvements over FIFO-Diffusion across all four dimensions. It is important to note that consistency remains a key and ongoing modeling goal in video generation. Even the latest commercial models, such as Sora2, still encounter consistency issues. Given the current state of the field, our substantial advances in both subject and background consistency are notable, and it is reasonable that consistency issues have not been fully resolved.
>
> ---
>
> It is our sincere hope that the above responses have addressed your remaining questions. We would appreciate it if you could reconsider your rating based on this additional information. Should you have any further questions, please feel free to let us know.
>
> ---
>
> [1]. Scalingnoise: Scaling inference-time search for generating infinite videos, Yang et al., arXiv 2503.16400.

---

### Official Review · Reviewer_cRYb · 2025-11-01

**Soundness:** 2
**Presentation:** 3
**Contribution:** 2
**Rating:** 4
**Confidence:** 4

**Summary:**

This paper addresses the challenge of extending pre-trained short video generation models to create coherent, long videos in a train-free manner. It identifies two major flaws in existing frame-level autoregressive frameworks, such as FIFO-Diffusion: 1) A significant "training-inference gap," as models trained on latents at a single noise level are forced during inference to process a queue of latents with a wide span of different noise levels, leading to artifacts. 2) A failure to model long-term dependencies, which results in poor temporal consistency as the video progresses.

To solve these issues, the authors propose MIGA, a method with two main contributions. First, it introduces a Two-Stage Training-Inference Alignment (TTA) mechanism. Stage 1 uses "zigzag iterative denoising," which groups latents by noise level (using a width of $L_{zig}$) to reduce the noise span seen by the model. Stage 2 completes the process by performing a unified denoising step once all latents have reached the same intermediate noise level $\tau_{e-1}$, perfectly aligning with the model's training conditions. Second, MIGA employs a Dual Consistency Enhancement (DCE) mechanism. This includes a "self-reflection" approach to efficiently evaluate consistency on new, high-noise latents and trigger a corrective search if anomalies are detected, and a "long-range frame guidance" method that injects older, low-noise latents ($m_{guid}$) into the current processing window to enforce temporal coherence. Experiments show MIGA achieves state-of-the-art results on VBench and NarrLV benchmarks.

**Strengths:**

+ The paper tackles a well-known, difficult problem (long-term consistency in long video generation) from a novel and practical angle. While it builds on existing frame-level autoregressive (AR) frameworks like FIFO-Diffusion, its contributions are highly original. The "Two-Stage Training-Inference Alignment" (TTA) mechanism, particularly Stage 2 (Unified Denoising), is a very clever solution to the training-inference gap by forcing the final denoising steps to perfectly match the training distribution (noise span of 1). Furthermore, the "Dual Consistency Enhancement" (DCE) mechanism's "Self-Reflection" component is also very original. The insight to use noisy-latent self-similarity as a computationally cheap proxy for clean-latent consistency (as shown in Fig. 3) to trigger a test-time search is a significant and non-obvious contribution.

+ TThe methodology is well-motivated, and the two main components (TTA and DCE) directly address the two identified weaknesses of prior work (training-inference gap and long-term consistency). The experimental validation is thorough and convincing. The authors use two different foundation models (VideoCrafter2 and Wan2.1) and two standard benchmarks (VBench, NarrLV). The ablation study in Section 4.3 is exemplary; it methodically deconstructs the proposed system, providing strong evidence for the independent and combined contributions of TTA (Stage 1 and Stage 2) and DCE (Self-Reflection and Long-Range Guidance). The qualitative results, especially the step-by-step ablation in Figure 4, clearly visualize the impact of each component.

+ The paper is exceptionally well-written and easy to follow. The core limitations of the baseline (FIFO-Diffusion) are explained with intuition (training-inference gap) and illustrated well (Fig. 2a). The proposed MIGA solution is logically broken down into its TTA and DCE components, each of which is explained with clear diagrams (Fig. 2b, Fig. A1) and, in the appendix, detailed pseudocode. The connection between the observed problem and the proposed solution is direct and compelling.

+ Training-free methods for extending foundation models are of immense practical value, as they democratize access to powerful capabilities (like long video generation) without requiring massive computational resources for retraining. This paper provides a robust and well-engineered solution that demonstrably improves the state-of-the-art in a challenging domain. By solving key issues in AR frameworks, MIGA makes infinite-frame generation significantly more consistent and stable, pushing the boundaries of what can be achieved without additional training.

**Weaknesses:**

- "Infinite Generation" Claim vs. TTA Stage 2: There appears to be a contradiction between the paper's claim of "infinite-frame generation" with "constant memory" (as inherited from FIFO-Diffusion) and the mechanics of the proposed TTA Stage 2. As described in Section 3.2 and Algorithm 5 (lines 16-26), Stage 2 collects all N partially denoised latents (where N is the total length of the final video) into a queue Q_gen. It then performs unified denoising steps on this entire queue. This implies that memory usage scales with N, the total number of frames to be generated. This is a fundamental departure from the streaming, constant-memory paradigm of FIFO-Diffusion, which generates one clean frame at a time. This re-introduces the very memory-scaling problem that AR frameworks were designed to solve. While the method is still "autoregressive" in Stage 1, the overall process seems to be for fixed-length (though long) video generation, not a truly "infinite" stream. This is a major weakness that needs to be clarified, as it affects the paper's core premise.

- Missing Latency/Throughput Analysis: The paper focuses entirely on quality improvements (VBench/NarrLV scores) but provides no analysis of the computational overhead. The baseline FIFO-Diffusion is already computationally intensive. The proposed MIGA adds two new sources of overhead: TTA (specifically Stage 2, which processes all N frames e-1 times) and DCE (which involves a test-time search, as acknowledged in Appendix B.3). While Appendix B.3 discusses the cost, it provides no concrete numbers. A key part of evaluating a new method is understanding its trade-offs. How much slower is MIGA than FIFO-Diffusion? Is it 1.1x, 2x, or 10x slower? Without a "Latency vs. Quality" comparison (e.g., in Table 1), it's hard to judge the practical utility of the method. The gains in consistency may not be worth a massive drop in throughput.

**Questions:**

- Memory Scaling of TTA Stage 2: Could the authors please clarify the apparent contradiction regarding the "infinite" and "constant memory" claims? Algorithm 5 suggests memory scales with N (total frames). The authors may explain the memory-management mechanism of Stage 2 that maintains constant memory.

- Practical Latency Overhead: What is the wall-clock latency (or throughput in frames/sec) of the full MIGA (TTA+DCE) method compared to the baseline FIFO-Diffusion when generating a video of a fixed length (e.g., 128 or 161 frames, as in Table 1)? A concrete comparison is needed to evaluate the practical cost-benefit trade-off of the proposed mechanisms.

- Effectiveness of "Zigzag" Denoising (TTA Stage 1): The "zigzag" denoising in Stage 1 is proposed to "proactively narrow the noise span". However, any sliding window of size f_0 (e.g., f_0=16) would still seem to cover at least two different noise levels (e.g., ...2, 2, 2, 3, 3...), giving a span of 2. The baseline FIFO-Diffusion has a span of f_0. Is the improvement (which is confirmed in Table 6) really just from going from a span of f_0 to a span of 2? Or is there another factor at play?

- Robustness of Self-Reflection Proxy: The use of noisy-latent similarity as a proxy for clean-latent consistency (Fig. 3) is a key insight for the "Self-Reflection" component. How sensitive is this? The paper mentions it works even at high noise levels (e.g., 40/50). Does this proxy become more or less reliable at different noise levels? How was the judgment index f_judg (the noise step at which to perform this check) chosen?

---

> ### Author Response · Authors · 2025-11-21
> **Response to Reviewer cRYb (Part 1 of 2)**
>
> **Dear Reviewer cRYb,**
>
> We sincerely appreciate your time and effort in reviewing our work. We are encouraged by your recognition of our novel, practical, and well-motivated approach; our original, clever, significant, and non-obvious methodological contributions; the thorough and convincing experimental validation that demonstrates the state-of-the-art performance of MIGA; and the exemplary, detailed ablation studies confirming the effectiveness of each module. We are also grateful for your positive remarks regarding the well-written and easy-to-follow presentation of our manuscript, as well as the inclusion of clear diagrams and detailed pseudocode.
>
> We have carefully considered your concerns, which include many valuable suggestions for improvement. After targeted analysis and revision, we hope the following responses will address your concerns.
>
> ---
>
> ### **Q1: Analysis of Infinite Frame Generation and the Impact of TTA Stage 2 on Memory Overhead**
>
> Thanks for your attention to MIGA's memory overhead. We provide the following clarifications regarding the infinite frame generation capability of MIGA and the impact of TTA Stage 2 on memory consumption:
>
> 1. **Infinite Frame Generation in FIFO-Diffusion:** As you pointed out, the core factor enabling infinite frame generation in video generation models is whether increasing the number of generated frames results in significant memory overhead. FIFO-Diffusion achieves frame-level autoregressive generation by maintaining a queue of frames. Since the number of frames in the queue typically exceeds what the foundation model can process in a single inference, FIFO-Diffusion relies on a sliding window denoising approach. Notably, during infinite frame generation, FIFO-Diffusion incurs a fixed memory overhead, which is primarily attributed to the denoising of the foundation model within the local sliding window. The queue length only affects the number of sliding denoising operations but has minimal impact on overall memory usage.
>
> 2. **Infinite Frame Generation in MIGA:** Based on the above, although the two-stage design in our TTA mechanism influence the queue length, the number of frames processed by the foundation model in each sliding window remains constant. Therefore, MIGA does not incur significant memory overhead as the number of generated frames increases. This ensures that MIGA, like FIFO-Diffusion, supports infinite frame generation.
>
> 3. **Quantitative Analysis:** We conducted a quantitative analysis of peak memory consumption (MiB) for VideoCrafter2-based MIGA under varying generation settings and numbers of frames. In the table below, "#1" represents the ablation setting without Stage 2. For reference, we also report the memory overhead for the foundation model (VideoCrafter2) during short-term inference, i.e., 9919 MiB. The values in parentheses indicate the proportion of additional memory consumption relative to VideoCrafter2. The results show:
>      - Memory overhead is not affected by the use of Stage 2 under different frame counts;
>      - Memory usage increases gradually as the number of generated frames grows, due to the requirement to store certain intermediate variables. However, the additional overhead remains minimal and acceptable.
>
> | # | 100   | 500   | 1000  | 1500  | 2000  |
> |---|-------|-------|-------|-------|-------|
> | 1 | MIGA w/o Stage 2       | 9929 (+0.10%) | 9945 (+0.26%) | 9965 (+0.46%) | 9985 (+0.66%) | 10005 (+0.86%) |
> | 2 | MIGA                   | 9929 (+0.10%) | 9945 (+0.26%) | 9965 (+0.46%) | 9985 (+0.66%) | 10005 (+0.86%) |
>
> ---
>
> ### **Q2: Analysis of Computational Overhead and Performance Gains**
>
> Thanks for your constructive suggestions. We conducted a comparative analysis of computational efficiency and performance for both FIFO-Diffusion and our MIGA on the same GPU device. As the Self-Reflection method within our Dual Consistency Enhancement (DCE) mechanism serves as a test-time scaling technique, it intentionally increases computational overhead to achieve higher performance. Thus, we discuss it separately.
>
> As shown in the table below, we report the models' computational efficiency (average time to generate one frame, $M_t$) and performance (Overall Score, O.S., on VBench) across different settings. The results indicate that, without the DCE mechanism, MIGA exhibits similar computational efficiency to FIFO-Diffusion (+0.05s), while delivering substantially better performance (+1.73). Furthermore, the addition of DCE further improves performance at the cost of increased computational overhead.
>
> | # | Setting |$M_t$  | O.S.   |
> |---|--------|--------|-------|
> | 1 | FIFO-Diffusion     | 7.48s  | 95.02  |
> | 2 | MIGA w/o DCE       | 7.53s  | 96.75  |
> | 3 | MIGA               | 9.16s  | 97.82  |

---

> > ### Author Response · Authors · 2025-11-21
> > **Response to Reviewer cRYb (Part 2 of 2)**
> >
> > ### **Q3: Effectiveness of TTA Stage 1**
> >
> > For your concerns regarding the details of TTA Stage 1, we provide the following clarifications:
> >
> > 1. **Effectiveness of Reduced Noise Span:** As shown in Table 4 of the paper, we analyze the impact of Zigzag width $L_{zig}$. A larger $L_{zig}$ corresponds to a smaller noise span. Notably, this ablation experiment varies only the value of $L_{zig}$. The results indicate that as $L_{zig}$ increases (i.e., as the noise span decreases), model performance improves initially and then plateaus. This suggests that reducing the noise span contributes to some performance gains.
> >
> > 2. **Necessity of Aggressively Reducing Noise Span:** As indicated above, the performance improvement from reducing noise span is bounded. Furthermore, due to the inherent frame-level autoregressive mechanism, the maintained queue must cover a certain noise span. During overlapping sliding-window denoising, it is inevitable that inputs with non-zero noise span are encountered. In Table 5, we investigate the scenario where the noise span is eliminated entirely (i.e., no longer following the autoregressive generation paradigm), and observe a significant drop in performance (see Section 4.3 "Study on TTA" for details). This indicates that aggressively minimizing the noise span is unnecessary.
> >
> > ---
> >
> > ### **Q4: Robustness of our Self-Reflection Proxy**
> >
> > Thanks for your interest in the implementation of our Self-Reflection proxy. As illustrated in Figure 3 (d), we use the consistency curve computed from fully denoised videos as a reference and report the correlation coefficients between this reference and the curves obtained at different denoising steps. A higher correlation indicates that consistency curves evaluated at such noise levels can reliably reflect true consistency issues, enabling consistency analysis in the early stages of denoising.
> >
> > Our results show that although the correlation coefficient decreases as the noise level increases, it remains consistently high across a wide range of noise intensities. This demonstrates that our metric is robust, meaning that $f_{judg}$ provides similar consistency evaluation ability within a certain range of values. It is worth noting that the correlation drops significantly at very high noise levels. To balance evaluation quality and computational cost, we recommend selecting $f_{judg}$ at a noise intensity slightly lower than the maximum Gaussian noise level.
> >
> > ---
> >
> > We hope that the above responses can address your concerns and would be grateful if you could reconsider your rating. If you have any additional feedback or require further clarification, please do not hesitate to let us know.

---

### Meta-Review · Area_Chair_VukM · 2025-12-03

**Summary:**

Based on the reviewers’ feedback and my own reading of the paper, the overall quality still needs improvement. While the paper proposes improvements to long-video generation through techniques like Training-Inference Alignment (TTA) and Dual Consistency Enhancement (DCE), it suffers from some flaws that undermine its core claims and novelty. We regret to inform you that this paper has not been accepted for this year’s conference. We hope the authors can address the relevant issues in subsequent revisions and achieve acceptance in future submissions.

**Reviewer Concerns:**

The authors' rebuttal demonstrates the state-of-the-art performance of MIGA, along with exemplary and detailed ablation studies that confirm the effectiveness and solid gains provided by each module. But the novelty of MIGA compared to prior work has not been well resolved.

**Reviewer Scores:**

The novelty of MIGA compared to prior work requires further elaboration.

---

### Decision · Program_Chairs · 2026-01-26

Reject